# Evolution of winter precipitation in the Nile-River watershed since the last glacial

Vera D. Meyer[1*], Jürgen Pätzold[1], Gesine Mollenhauer[1,2], Isla S. Castañeda[3], Stefan Schouten[4], Enno Schefuß[1*]

[1]MARUM – Center for Environmental Sciences, University of Bremen, Bremen, 28359, Germany
[2]Alfred Wegener Institute, Helmholtz Centre for Polar and Marine Research, Bremerhaven, 27570, Germany
[3]Department of Earth, Geographic and Climate Science, University of Massachusetts Amherst, Amherst, MA USA
[4]NIOZ Royal Netherlands Institute for Sea Research, Department of Marine Microbiology and Biogeochemistry, 1790 AB Den Burg, Netherlands

*Correspondence to*: Vera D. Meyer (vmeyer@marum.de), Enno Schefuß (eschefuss@marum.de)

**Abstract.** Between 14.5 and 5 ka BP, the Sahara was vegetated owing to a wet climate during the African Humid Period (AHP). However, the climatic factors sustaining the "Green Sahara" are still a matter of debate. Particularly the role of winter precipitation is poorly understood. Using the stable hydrogen isotopic composition (δD) of high molecular weight (HMW) *n*-alkanoic acids in a marine sediment core from the Eastern Mediterranean (EM), we provide a continuous record for winter precipitation in the Nile-River delta spanning the past 18 ka. Pairing the data with regional δD records from HMW *n*-alkanes, we show that HMW *n*-alkanoic acids constantly derive from the delta while the HMW *n*-alkanes also receive significant contributions from the headwaters between ~15-1 ka BP due to enhanced fluvial runoff. This enables us to reconstruct the evolution of Mediterranean (winter) and monsoonal (summer) rainfall in the Nile River watershed in parallel. Heinrich Stadial 1 (HS1) evolved in two phases with a dry episode between ~17.5-16.0 ka BP followed by wet conditions between ~16-14.5 ka BP owing to movements of the Atlantic storm track. Winter rainfall enhanced substantially between 11-6 ka BP, lagging behind the intensification of the summer monsoon by ca. 3 ka. Heavy winter rainfall resulted from a southern position of the Atlantic storm track combined with elevated sea-surface temperatures in the EM reinforcing local cyclogenesis. We show that during the "Green Sahara", monsoon precipitation and Mediterranean winter rainfall were simultaneously enhanced and infer that the winter-rainfall zone extended southwards delivering moisture to the Sahara. Our findings corroborate recent hypotheses suggesting southward extended winter rains were a crucial addition to the northward displacement of the summer monsoon in helping to sustain a "Green Sahara".

## 1 Introduction

North Africa underwent dramatic oscillations between dry and wet climate states during glacial-interglacial cycles (e.g., deMenocal et al., 2000a; Menviel et al., 2021; Larrasoaña et al., 2013; Ziegler et al., 2010). The last wet phase, known as the "African Humid Period" (AHP), occurred between 14.5-5 ka BP reaching its climax between 11-6 ka BP (e.g. deMenocal et

al., 2000a; Shannahan et al., 2015; Tierney et al., 2017; Dupont and Schefuß. 2018). Humid conditions in North Africa transformed the formerly barren, hyperarid Sahara Desert into a fertile "Green Sahara" (~11.5-5 ka BP; Kuper and Kröpelin, 2006) where savannah, lakes, rivers and wetlands existed allowing human settlements in the Sahara (Kuper and Kröpelin,

2006; Quade et al., 2018; Larrasoaña et al., 2013; Jolly et al., 1998). Although intensively studied, the drivers and spatio-temporal extent of the AHP are still a matter of debate (Lüning and Vahrenholt, 2019; Kutzbach et al., 2014; Otto-Bliesner et al., 2014; Menviel et al., 2021; Tierney et al., 2017; Sha et al., 2019; Cheddadi et al., 2021). There is consensus that the AHP initiated in response to insolation forcing, which intensified the African summer monsoon and shifted the Intertropical Convergence Zone and the African rainbelt northward (Menviel et al., 2021; Lüning and Vahrenholt, 2019; Pausata et al.,

2016; Braconnot et al., 2007; Claussen et al., 2017; deMenocal et al., 2006b). Next to orbital forcing positive, non-linear feedbacks from the land surface amplified the climatic changes (e.g. Chandan et al., 2020; Pausata et al., 2016; 2020).

Controversy exists on the termination of the AHP (Kuper and Kröpelin, 2006; Shannahan et al., 2015; Schefuß et al., 2005; Costa et al, 2014; Blanchet et al., 2014; deMenocal et al., 2000a; Tierney and deMenocal, 2013; Ménot et al., 2020; Tierney et al., 2008; Berke et al., 2012; Weldeab et al., 2014; Junginger et al., 2014; Castañeda et al., 2016a; deMenocal, 2015; Collins

et al., 2017) as well as on the climatic processes sustaining a vegetated Sahara (Cheddadi et al., 2021, Kutzbach et al., 2014; Alpert et al., 2006; Chandan et al., 2020; Braconnot et al., 2007; Hopcroft et al., 2017; Claussen et al., 2017; Sha et al., 2019; Tierney et al., 2017). Paleoclimate records suggest both a gradual as well as an abrupt ending of the AHP and indicate that the AHP terminated earlier in the North than in the South (Kuper and Kröpelin, 2006; Shannahan et al., 2015; deMenocal, 2015). For a long time, most studies focused on the northward extension of the summer monsoon seeking to explain the AHP and the

Green Sahara (deMenocal et al., 2006b; Braconnot et al., 2007; Menviel et al., 2021; deMenocal et al., 2000b; Shannahan et al., 2015; Sha et al., 2019; Tierney et al., 2017). However, reproducing the AHP and the "Green Sahara" has challenged climate modellers for a long time because of inconsistencies between model results and proxy data, a problem that has not been overcome yet (Chandan and Peltier., 2020; Braconnot et al., 2007; Hopcroft et al., 2017; Perez-Sanz et al., 2014). Climate models commonly suggest a northward extension of the summer monsoon (Chandan and Peltier., 2020; Braconnot et al., 2007;

Hopcroft et al., 2017; Perez-Sanz et al., 2014) but in northern Africa these models underestimate precipitation amount inferred from the pollen-based reconstructions (e.g. Bartlein et al., 2011; Hély et al., 2014) and isotopic compositions of biomarkers and speleothems (Tierney et al., 2017; Sha et al., 2019). The representation of positive feedbacks from the land surface on the monsoon may partly account for the discrepancies as these feedbacks substantially amplify the response of the monsoon to orbital forcing, pushing the monsoon further north than orbital forcing alone (Chandan and Peltier, 2020; Pausata et al., 2016).

Nevertheless, inconsistencies with proxy data remain (e.g. Chandan and Peltier, 2020; Pausata et al., 2016). The simulated precipitation in northern Africa is still too low compared to proxy data while it overestimates the pollen-based inferences between 10-15°N (Pausata et al., 2016; Chandan and Peltier, 2020; Bartlein et al., 2011) but are compatible with rainfall estimates based on leaf-wax isotopic compositions (Tierney et al., 2017). Given these inconsistencies contributions from additional moisture sources such as the Mediterranean winter rains increasingly receive attention to explain the green Sahara

(Kutzbach et al., 2014; Cheddadi et al., 2021; Tierney et al., 2017).

Recently, Cheddadi et al. (2021) suggested that intensified Mediterranean winter rainfall and a southward extension of the winter rainfall zone into the Sahara may have delivered the additional moisture needed for sustaining a green Sahara by decreasing rainfall seasonality. Beforehand, Tierney et al. (2017) had already invoked additional winter precipitation to fully explain their observed amplitude in precipitation records from northwestern Africa. A model-based investigation of the response of Mediterranean winter rainfall confirms intensified precipitation in northern Africa and the Middle East during the AHP in response to orbital forcing (Kutzbach et al., 2014).

Unfortunately, the glacial-to-Holocene development of winter precipitation in northeastern Africa remains elusive given the scarcity of proxy records. To improve the understanding of how winter and summer precipitation evolved around the AHP, continuous records for precipitation are required to robustly investigate spatial variations in rainfall across North Africa. Constraints on the hydroclimatic development in the southeastern Mediterranean come from Earth System Models and various paleoenvironmental archives from the Asian borderlands of the Levantine Basin (Langgut et al., 2016; Tierney et al., 2022: Stein, 2001; Ludwig and Hochmann, 2022; Bar-Mathews et al., 1997; Goldsmith et al., 2017; Enzel et al., 2008; Cheng et al., 2015; Miebach et al., 2019). While there is consensus that during the Last Glacial Maximum (LGM) the eastern Mediterranean realm was characterized by a more positive moisture balance (Stein, 2001; Goldsmith et al., 2017; Ludwig and Hochmann, 2022), the effects from changes in evaporation and precipitation on effective moisture are still debated. Some authors proposed increased precipitation (Enzel et al., 2008; Stein et al., 1997; Cheng et al., 2015) while others suggested reduced precipitation (Ludwig and Hochmann, 2022; Bar-Mathews et al., 1997; Langgut et al., 2021). While the sign of rainfall changes is ambiguous, there is a broad consensus among proxy data and model results that evaporation was lower under cold glacial conditions (Stein, 2001; Tierney et al., 2022; Ludwig and Hochmann, 2022; Goldsmith et al., 2017). These uncertainties in terms of rainfall ask for more studies addressing the hydroclimate development in the Eastern Mediterranean since the last glacial.

A key region to study northeastern-African climate change is the Nile-River basin that extends over 3 million km$^2$ (Figure 1). Currently the Nile basin is influenced by monsoonal summer rains south of the Sahara Desert and Mediterranean winter rainfall in the delta region. Furthermore, it is of societal relevance to address the climatic history of the Nile River catchment because the river is the lifeline of Egypt, providing fertile ground and drinking water to millions of people. It also played a vital role in the rise and demise of ancient Egyptian civilizations (e.g. Zaki et al., 2021). In northeastern Africa and the borderlands of the southern Levantine basin, socioeconomic impacts due to increasing aridity are expected with future warming. Continuous archives for precipitation are predominantly found in the headwaters of the Nile River and further south (Lakes Victoria, Tana, Tanganyika; Figure 1; Berke et al., 2012; Costa et al., 2014; Tierney et al., 2008; 2010). In the northern part of the catchment where the hyperarid Sahara Desert extends, continuous records for precipitation are sparse as sedimentary sequences from the deglaciation and the Holocene, such as lacustrine deposits, were subject to strong wind erosion during arid periods (Hamdan et al., 2016; Hamdan and Lucarini, 2013). Sediment cores from the Levantine Basin (Figure 1) are used to reconstruct environmental changes in the Nile-River watershed. These records are commonly considered as integrators of the entire

catchment and are mostly interpreted to reflect monsoonal rainfall variability (Castañeda et al., 2016a; Revel et al., 2015; Ménot et al., 2020, Blanchet et al., 2014). However, the Nile crosses several climate regimes that drastically differ in precipitation amount and seasonality: tropical conditions in the headwaters, hyperarid desert in its central part, and Mediterranean winter rainfall in the delta region. For climate reconstructions, it is crucial to address these climate zones separately to identify latitudinal differences and to understand how monsoonal (summer) and Mediterranean (winter)

precipitation evolved around the AHP.

Here, we provide a new hydroclimate record based on the stable hydrogen isotopic composition ($\delta^2$H or also often termed: $\delta$D where D stands for deuterium) of HMW $n$-alkanoic acids. HMW $n$-alkanoic acids are major components of epicuticular leaf-waxes of higher plants (Eglinton and Hamilton, 1967). The record is obtained from marine sediment core GeoB7702-3 from

the Eastern Mediterranean Sea (EM; Figure 1). $\delta$D of leaf-wax lipids ($\delta D_{wax}$) is a powerful means to reconstruct past hydrological changes (e.g. Sachse et al., 2012) and has been successfully applied to infer hydroclimate variability across Africa (Schefuß et al., 2005; Tierney et al., 2008; 2017; Berke et al., 2012; Costa et al., 2014; Collins et al., 2013; Castañeda et al., 2016a; Konecky et al., 2011). We infer that $\delta D_{wax}$ of our HMW $n$-alkanoic acids records winter precipitation in the Nile delta region. By comparison to existing $\delta D_{wax}$ records based on HMW $n$-alkanes from the same core (Castañeda et al., 2016a) and

HMW $n$-alkanoic-acid based $\delta D_{wax}$ records from the headwaters (Berke et al., 2012; Costa et al., 2014; Tierney et al., 2008), we are able to examine hydroclimate variability in the southern and northern sections of the Nile-River catchment and shed light on the interplay of Mediterranean (winter) and monsoonal (summer) rainfall changes in northeastern Africa around the AHP.

## 2 Study Area

Core GeoB7702-3 was recovered from the southeastern Levantine Basin off Israel (Figure 1) (Pätzold et al., 2003). The core receives terrigenous material from the Nile River as the suspension load is transported eastward along the continental margin due to the anticlockwise direction of surface currents and eddies (e.g. Weldeab et al., 2002). This makes site GeoB7702-3 a suitable archive for environmental changes in the Nile-River basin (Castañeda et al., 2010a; 2016a).

The Nile River is the longest river in the world extending over 34° of latitude (Figure 1). The catchment spans from Equatorial

Africa to the Mediterranean coast draining Uganda, Ethiopia, South Sudan, Sudan and Egypt (Figure 1). The river consists of three major tributaries, i.e. the White Nile (sourced from Lake Victoria), the Blue Nile (sourced from Lake Tana) and the Atbara Nile (source is situated north of Lake Tana). The confluence of the three tributaries forms the Main Nile (Figure 1). For the following discussion, we define three sub-catchments of the watershed as follows: upper catchment (headwaters in Ethiopia and Uganda; 4°S to 15°N), middle catchment (Sahara, Sudan and Egypt; 15-30°N) and lower catchment (Nile delta;

30-31°N). On its way to the North the Nile River crosses different climate zones and vegetation regimes. The climate zones

encompass Mediterranean climate in the delta area, the hyperarid Sahara Desert, the semiarid Sahel zone and a wet, tropical climate in Ethiopia and Uganda (e.g. Korecha and Barnston, 2007). In the lower and middle catchment, rainfall mainly occurs

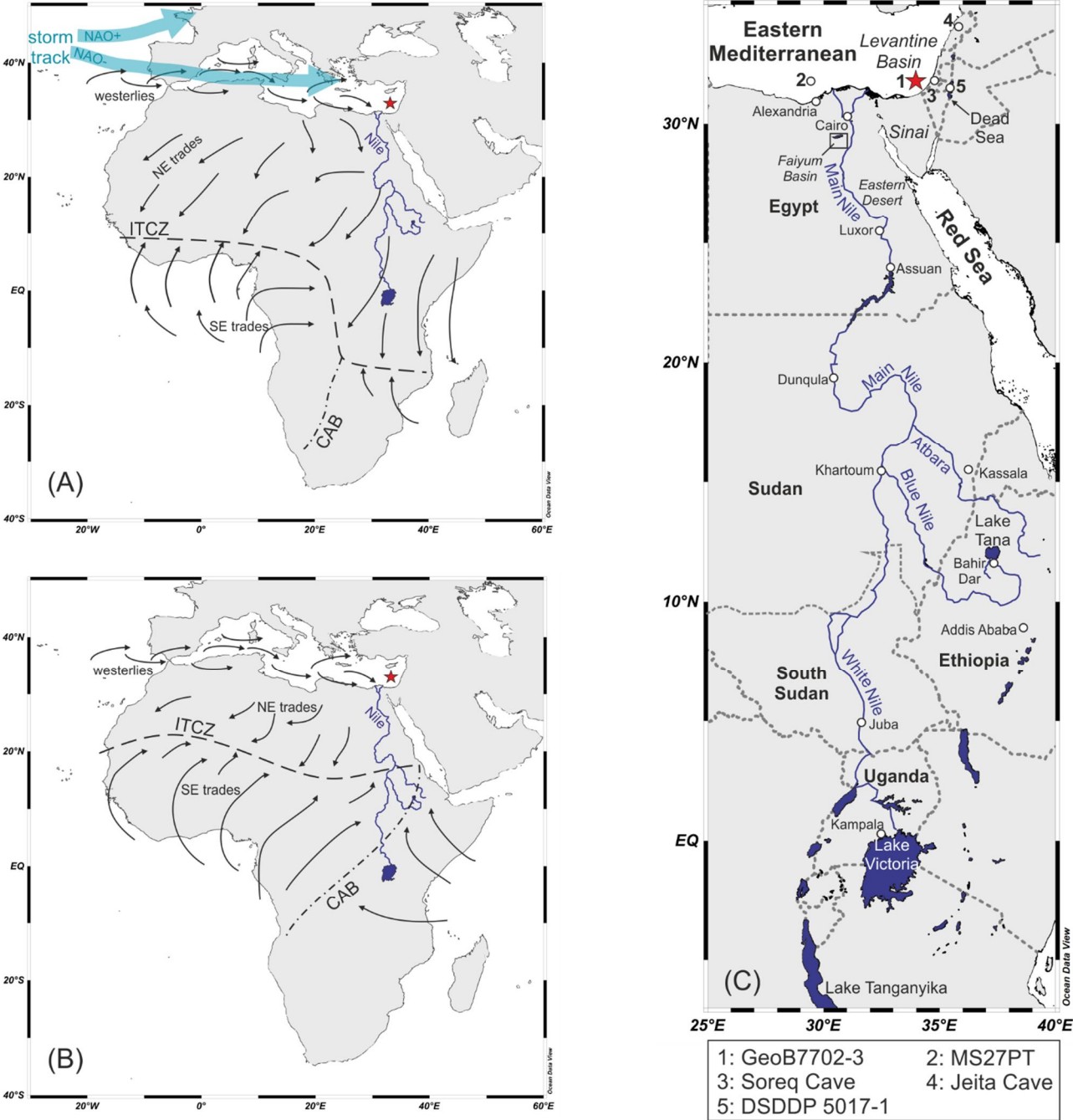

**Figure 1: General wind patterns over Africa during winter (A) and summer (B). The star marks site GeoB7702-3. The modern**
**positions of the ITCZ during January (A) and July/August (B) are illustrated. The Congo Air Boundary (CAB), which separates**
**Atlantic and Indian air masses over equatorial Africa. In (A) the positions of the North Atlantic storm track are sketched for positive**

and negative phases of the North Atlantic Oscillation (NAO). (C) Detailed map of the Nile-River watershed. Site GeoB7702-3 is marked by a red star. Other locations mentioned in the text are indicated by circles. The maps were created using Ocean Data View 5.6.3 (Schlitzer, 2006).

during winter. In the Nile delta annual rainfall spans from 181 mm/yr at the coast (Alexandria; https://en.climate-data.org) to 18 mm/yr at its southern edge (Cairo; https://en.climate-data.org). Most rain falls between October and March. The Sahara usually does not receive any rainfall (e.g. Luxor; https://en.climate-data.org). The upper section of the watershed is characterized by heavy rainfall (e.g. about 1874 mm/yr annually in Addis Ababa/Ethiopia; about 1747 mm/yr in Kampala/Uganda; https://en.climate-data.org). In Ethiopia, most rain falls between June and September (https://en.climate-

data.org). Uganda receives year-round precipitation with rainfall maxima during March till May and September till November (https://en.climate-data.org).

Precipitation in the upper Nile River watershed is mainly determined by the West African Monsoon, and thus related to the seasonal migration of the Intertropical Convergence Zone (ITCZ; Figure 1). During the summer months, the seasonal northward movement transports moisture-laden air to southern northeastern Africa (up to ~15°N). Additionally, the Congo Air

Boundary controls the relative contribution of moisture from the Gulf of Guinea, i.e. the Atlantic Ocean, and Indian Ocean to the headwaters (Figure 1) (Camberlin, 2009). North of 15°N the catchment is predominantly under the influence of the westerlies receiving moisture from the Atlantic Ocean and the Mediterranean Sea, the Arabian Peninsula and the Red Sea (Figure 1) (Viste and Sortberg, 2013). The Mediterranean climate is characterized by dry summers and wet winters. Winter precipitation is governed by cyclogenesis over the eastern Mediterranean, so called "Cyprus Lows". The cyclones form when

relatively cold European air encounters the warm Mediterranean Sea during southward advection (Alpert et al., 1990). Next to the Cyprus Lows, the Red Sea Trough is another local atmospheric circulation pattern controlling winter rainfall. It is commonly associated with warm and dry conditions but variations in its position can cause heavy precipitation events (Tsvieli and Zangvil, 2005; Krichak et al., 2012). The winter climate is largely dependent to teleconnections with the North Atlantic Oscillation (NAO), which forms a constituent of the Arctic Oscillation. Modulating the positions of the Atlantic and

Mediterranean storm tracks (Figure 1), the NAO exerts strong control over moisture delivery to the Mediterranean borderlands. Warm and wet winters in the Mediterranean are generally associated with negative NAO-states while cold and dry winters occur during positive phases (Eshel and Farrel, 2000).

## 3 Material and Methods

### 3.1 Core Material and Chronology

Gravity core GeoB7702-3 was recovered from the continental slope off Israel at 562 m water depth during RV Meteor cruise M52/2 in 2002 (Pätzold et al., 2003). Prior to sampling the core was stored at 4°C. Age control for this core was previously established by Castañeda et al. (2010a) and is based on accelerator mass spectrometry (AMS) radiocarbon dates of planktic foraminifera. We refined the age model using up-to-date calibration curves and nine additional AMS dates (Table 1). To isolate foraminifera the samples were wet-sieved and specimens of the foraminifera *Globigerinoides ruber* alba were hand-picked

from the 150-63 μm fraction. In cases where the abundance of *G. ruber* was insufficient, we mixed planktonic foraminifera species to obtain enough material (Table 1). 290-810 μg carbon was dated at the MICADAS AMS-dating facility at the Alfred Wegener Institute, Helmholtz Centre for Polar and Marine Research (Bremerhaven, Germany) according to in-house protocols (Mollenhauer et al., 2021). The dates were combined with the data set of Castañeda et al. (2010a) to create the age-depth model. All radiocarbon dates are listed in Table 1. The BACON 2.5.8 software (Blaauw and Christen, 2011) was used for age-depth modelling. Radiocarbon ages were transferred into calendar ages based on the Marine20 calibration curve (Heaton et al., 2020). Today, the mean reservoir age offset (ΔR) in the Levantine Basin is -94 ±94yrs in relation to Marine20 (marine reservoir correction database; Reimer et al., 2001). BACON was run with a constant ΔR= -100 ±100 yrs, accordingly. Default settings were used for the prior, apart from the accumulation rate, which was set to 50 yrs/cm. BACON operated with 118 core-slices. The age-depth model for GeoB7702-3 and the prior distributions are plotted in Figure 2.

### 3.2 Lipid extraction, quantification and isotopic analysis

For biomarker analyses, a total of 51 samples were collected every 5-12 cm, providing a mean temporal resolution of ~350 years between samples. Target biomarkers of our study are HMW *n*-alkanes and HMW *n*-alkanoic acids and their stable isotope compositions. The lipid extraction and isotopic analysis were performed at MARUM - Centre for Marine Environmental Sciences (University of Bremen, Germany). The sediment samples were freeze-dried and afterwards homogenized using a mortar and pestle. The soluble organic matter was extracted from ~5 g of sediment using an Accelerated Solvent Extractor (ASE200). Extraction was performed with three cycles lasting 5 minutes each, using Dichloromethane (DCM):Methanol (9:1 v/v) at 100°C and 1000 psi. 19-Methyl-Arachidic Acid were added to the samples as internal standards prior to extraction. The total lipid extracts were saponified using potassium hydroxide (KOH). Neutral lipids and fatty acids were recovered using *n*-hexane and DCM, respectively. The fatty acids were methylated using methanol of known isotopic signature. The fatty-acid methyl esters (FAMEs) were cleaned by means of column chromatography. Columns consisted of 4 cm deactivated silica (1% $H_2O$ in *n*-hexane) and 0.5 cm sodium sulfate ($Na_2SO_4$) in a Pasteur pipette with 5 mm diameter. FAMEs were recovered using DCM:Hexane (2:1). FAMEs were quantified by gas chromatography and flame ionization detection (GC-FID) using a Thermo FOCUS GC. The GC was equipped with a Restek Rxi-5ms column. FAMES were analysed using a similar method as in Gensel et al. (2022). For the quantification of the FAMEs of HMW *n*-alkanoic acids the response factors of the target compounds were determined from an external standard mixture containing five HMW *n*-alkanoic acid homologues ($C_{24:0}$, $C_{26:0}$, $C_{28:0}$, $C_{30:0}$ and $C_{32:0}$) at a concentration of 50 ng/μl. The external standards were run every six samples. The relative standard deviation of the peak areas was <9%.

Isotope analysis of stable hydrogen (δD) of the HMW *n*-alkanoic acids was performed using gas chromatography coupled to isotope ratio mass spectrometry (GC-IRMS). We used a Thermo Trace GC coupled to a MAT253 MS. Isotope values were measured against calibrated reference gas ($H_2$). Values are reported in per mil relative to the VSMOW standards. A standard mixture consisting of 16 *n*-alkanes was run every sixth sample to monitor the performance of the system. For the δD analysis the accuracy and precision (mean deviation from offline values and the respective relative standard deviation, RSD) were

2.2‰ and 3.0‰, respectively. The instrument was operated only when the average absolute deviation from offline values was <5‰. The $H_3^+$ -factor was measured daily and was 5.6 ±0.1 throughout the measurement series. Replicate measurements of the samples yielded a standard deviation of 0.1-3.8‰ for δD. We report the δD-signatures of the $n$-$C_{26:0}$ and $n$-$C_{28:0}$ alkanoic acids as they are the most abundant homologues in our samples. The δD of the respective FAMEs were corrected for the bias introduced during the methylation process using isotope mass balance (hereafter $δD_{wax\ n\text{-alkanoic acid}}$). $δD_{wax\ n\text{-alkanoic acid}}$ was further corrected for deglacial changes in global ice-volume applying stacked data of oxygen isotopic compositions ($δ^{18}O$) of benthic foraminifera (L04-stack; Lisiecki and Raymo, 2005) and the approach described in Ruan et al. (2019). We additionally analyzed the stable carbon isotopic composition of HMW $n$-alkanoic acids ($δ^{13}C_{wax\ n\text{-alkanoic acids}}$) as described in the supplementary material (S1).

Prior to this study, a separate set of samples from GeoB7702-3 were extracted at the Royal Netherlands Institute for Sea Research (NIOZ) (Castañeda et al, 2010). As described in Castañeda et al. (2016), neutral lipids were split in apolar, keto and polar fractions by alumina oxide column chromatography with the following solvent mixtures: hexane:dichloromethane (9:1 v/v), hexane:DCM (1:1 v/v) and DCM:methanol (1:1 v/v). The apolar hydrocarbon fraction was further separated into saturated and unsaturated compounds using silica gel coated with silver nitrate ($AgNO_3$). Identification of $n$-alkanes was performed on a Thermo Finnigan Trace Gas Chromatograph (GC) Ultra coupled to Thermo Finnigan DSQ mass spectrometer (MS) using a CP Sil-5 fused silica capillary column (25 m×0.32 mm; film thickness 0.12 μm) with helium as the carrier gas. Mass scans were made from $m/z$ 50 to 800, with 3 scans per second and an ionization energy of 70 eV. The oven program started at 70°C, increased by 20°C min$^{-1}$ to 130°C, and subsequently by a rate of 4°C min$^{-1}$ until 320°C (held for 10 min). Quantification of $n$-alkanes was performed on an HP 6890 gas chromatograph (GC) using a 50 m CP Sil-5 column (0.32 mm diameter, film thickness 0.12 μm) and helium as the carrier gas, and with the same oven temperature program as for GC-MS. Compound concentrations were determined by relating chromatogram peak areas to the concentration of an internal standard of known concentration. Details regarding hydrogen and carbon isotopic analyses of the $n$-$C_{29}$ and $n$-$C_{31}$ alkanes are reported by Castañeda et al. (2016a).

Note that the set of samples taken for HMW $n$-alkanoic acids (this study) are at different sampling depths from the HMW $n$-alkanes (Castañeda et al., 2016a).

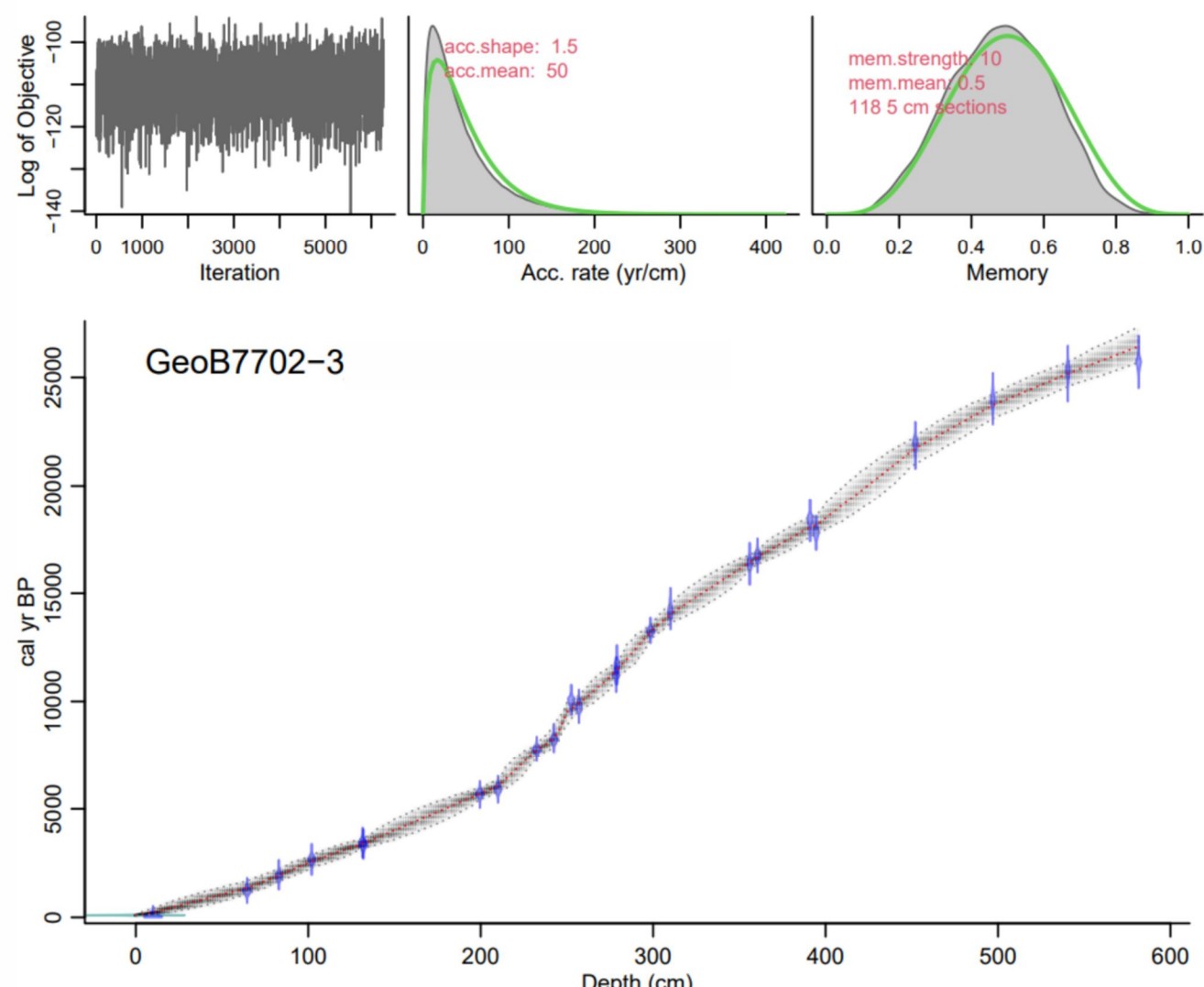

**Figure 2: Output graph for the age-depth modelling approach for core GeoB7702-3 using the BACON software (Blaauw and Christen, 2011). The model is based upon the radiocarbon dates of foraminifera given in Table 1. The small panels on top represent the Marcov Chain Monte Carlo iterations (left) and the distributions of the prior (green lines) and the posterior (grey histograms) for the accumulation rate (middle) and memory (right). The bottom panel illustrates the age-depth model. The blue dots depict the calibrated [14]C-ages. The grey stippled lines mark the 95% confidence intervals. The red line reflects the single best model based on the mean age for each depth. For more technical information the reader is referred to Blaauw and Christen (2011).**

Table 1: List of AMS-dates and corresponding calendar ages (cal. age) used to establish the revised age-depth model of GeoB7702-3. AMS dates were compiled from this study and Castañeda et al. (2010a; 2010b). Samples from this study were dated at the MICADAS-dating facility at Alfred Wegener Institute (Bremerhaven, Germany) while Castañeda et al. (2010a; 2010b) performed dating at the Leibnitz Laboratory for Radiometric Dating and Stable Isotope Research (University of Kiel, Germany). Our new age-depth model is based upon the median values of the calendar ages calculated.

| Depth [cm] | Sample label | Dated material | AMS date [14C a BP] | 2σ [±] | Cal. age min [a BP] | Cal. age max [a BP] | Cal. age median [cal a BP] | Cal. age mean [cal a BP] | Reference of AMS date |
|---|---|---|---|---|---|---|---|---|---|
| 0 | set* | - | 100* | - | 64 | 124 | 98 | 97 | this study |
| 10 | KIA25649 | *G. ruber* and *O. universa* | 245 | 30 | 123 | 415 | 221 | 231 | Castañeda et al. (2010b) |
| 64.5 | KIA25648 | *G. ruber* and *G. sacculifer* | 1725 | 25 | 1086 | 1682 | 1342 | 1356 | Castañeda et al. (2010b) |
| 81.5-84.5 | 6279.2.1 | *G. ruber* | 2340 | 26 | 1615 | 2285 | 1929 | 1933 | this study |
| 102 | KIA24619 | *G. ruber* | 2965 | 55 | 2288 | 2849 | 2580 | 2576 | Castañeda et al. (2010b) |
| 130-133 | 6281.1.1 | *G. ruber* | 3586 | 24 | 3113 | 3685 | 3400 | 3400 | this study |
| 132 | KIA24617 | *G. ruber* | 3500 | 35 | 3189 | 3643 | 3413 | 3416 | Castañeda et al. (2010b) |
| 198-201 | 6277.2.1 | *G. ruber* | 5399 | 25 | 5353 | 6014 | 5697 | 5697 | this study |
| 210 | KIA24616 | *G. ruber* | 5600 | 40 | 5764 | 6383 | 6036 | 6043 | Castañeda et al. (2010b) |
| 231-234 | 7307.1.1 | *G. ruber* | 7393 | 45 | 7242 | 8138 | 7724 | 7718 | this study |
| 242.5 | KIA25646 | *G. ruber* and *G. sacculifer* | 7845 | 40 | 8023 | 8866 | 8327 | 8358 | Castañeda et al. (2010b) |
| 251-254 | 6280.1.1 | *G. ruber* | 9309 | 28 | 9015 | 10110 | 9663 | 9652 | this study |
| 257 | KIA24613 | *G. ruber* | 9070 | 60 | 9564 | 10290 | 9934 | 9928 | Castañeda et al. (2010b) |
| 278-281 | 6276.2.1 | mixed planktonic foraminifera | 10144 | 34 | 11051 | 12047 | 11502 | 11508 | this study |
| 279 | KIA24612 | mixed planktonic foraminifera | 10470 | 70 | 11128 | 11839 | 11457 | 11463 | Castañeda et al. (2010b) |
| 297-300 | 6275.2.1 | *G. ruber* | 11827 | 32 | 12688 | 13729 | 13215 | 13213 | this study |
| 310 | KIA24611 | mixed planktonic foraminifera | 12580 | 80 | 13593 | 14509 | 14035 | 14037 | Castañeda et al. (2010b) |
| 356 | KIA24609 | *G. ruber* | 14130 | 100 | 16073 | 16781 | 16432 | 16431 | Castañeda et al. (2010b) |
| 359-362 | 6274.2.1 | *G. ruber* | 14420 | 37 | 16264 | 17074 | 16666 | 16664 | this study |
| 391 | KIA24608 | *G. ruber* | 15830 | 120 | 17610 | 18515 | 18036 | 18048 | Castañeda et al. (2010b) |
| 393-396 | 6262.2.1 | *G. ruber* | 15275 | 39 | 17690 | 18728 | 18151 | 18176 | this study |
| 452 | KIA25652 | *G. ruber* | 18810 | 150 | 21025 | 22260 | 21750 | 21717 | Castañeda et al. (2010b) |
| 497 | KIA24605 | *G. ruber* | 20660 | 180 | 23106 | 24215 | 23747 | 23716 | Castañeda et al. (2010b) |
| 540.5 | KIA24604 | *G. ruber* | 21840 | 220 | 24579 | 25711 | 25192 | 25179 | Castañeda et al. (2010b) |
| 581.5 | KIA25653 | mixed planktonic foraminifera | 22230 | 190 | 25715 | 27340 | 26397 | 26436 | Castañeda et al. (2010b) |

* the core-top age was defined as 100 calendar ages in order to guarantee BACON calculates positive dates for the depths above the first AMS-date (i.e. 10 cm).

## 4. Results

Concentrations of HMW $n$-alkanoic acids and HMW $n$-alkanes are shown in Figure 3. Concentrations of HMW $n$-alkanoic acids are slightly lower during the early deglaciation (between 18-17 ka BP) than during the late Holocene (Figure 3c). At ~16.5 ka BP they decrease to a minimum. From 14.6 ka BP onwards, the contents progressively increase in two steps culminating in a maximum at 10 ka BP (Figure 3c). Afterwards they decrease until the middle Holocene and increase again from 2 ka BP onwards. Concentrations of HMW $n$-alkanes show a similar development but differences are evident. The concentrations of HMW $n$-alkanes share the two-step increase starting at 14.6 ka BP and subsequent decline as well as the rise during the late Holocene (Figure 3b,c). However, while concentrations of HMW $n$-alkanoic acids are lowest during the early deglaciation (18-14.6 ka BP) the abundances of HMW $n$-alkanes are higher or similar to Holocene values (Figure 3b). Between ~17.5-16.5 ka BP concentrations of HMW $n$-alkanes are prominently elevated reaching their maximum at that time. By contrast, HMW $n$-alkanoic acids reach maximum contentrations around 10 ka BP (Figure 3b,c).

$\delta D_{wax\ n\text{-alkanoic acid}}$ are given in Figure 4. $\delta D_{wax\ n\text{-alkanoic acid}}$ range between -104 and -153‰. The $n$-$C_{26:0}$ and $n$-$C_{28:0}$ alkanoic acids behave similarly regarding $\delta D$-values and trends (Figure 4d; green and orange lines). In general, $\delta D_{wax\ n\text{-alkanoic acid}}$ is higher during the deglacial than during the Holocene (Figure 4d). Two episodes of remarkable change in $\delta D_{wax\ n\text{-alkanoic acid}}$ are evident in the record, i.e. between ~17.5-14.5 ka BP and between ~11-6 ka BP. At 17.5 ka BP $\delta D_{wax\ n\text{-alkanoic acid}}$ rapidly increase and reach maximal values at 17.0 ka BP. Afterwards, at ~16 ka BP, $\delta D_{wax\ n\text{-alkanoic acid}}$ decrease again reaching values similar to the late Holocene. At 14.5 ka BP $\delta D_{wax\ n\text{-alkanoic acid}}$ slightly increase again and then remain relatively constant until a striking minimum is registered between 10.0-6.0 ka BP. The maximum begins and terminates abruptly (where abruptly is defined as within a few hundred years). At this time HMW $n$-alkanoic acids are depleted by 34 ‰, relative to the values found at 12 ka BP (Figure 4d). After the minimum there is little variability throughout the late Holocene.

Interestingly, comparing our $\delta D_{wax\ n\text{-alkanoic acid}}$ with previously published $\delta D_{wax}$ of HMW $n$-alkanes (another leaf-wax lipid; $\delta D_{wax\ n\text{-alkanes}}$) from the same core (Figure 4d, blue line Castañeda et al., 2016a) reveals distinct discrepancies. While $\delta D_{wax\ n\text{-alkanoic acid}}$ and $\delta D_{wax\ n\text{-alkanes}}$ from GeoB7702-3 are similar before 15 ka BP and at 1 ka BP, they differ in the long-term trend of the glacial-to-Holocene evolution. The amplitude of the long-term development in the $\delta D_{wax\ n\text{-alkanes}}$ is larger than in the $\delta D_{wax\ n\text{-alkanoic acid}}$, which is why the $\delta D_{wax\ n\text{-alkanes}}$ are more negative than the $\delta D_{wax\ n\text{-alkanoic acid}}$ (up to 25 ‰; Figure 4d). This difference in amplitude mainly stems from a progressive decrease in $\delta D_{wax\ n\text{-alkanes}}$ from 15 ka BP onwards and from a progressive increase after ~5 ka BP (Figure 4d) this is not recorded by $\delta D_{wax\ n\text{-alkanoic acids}}$. In contrast to the $\delta D_{wax\ n\text{-alkanes}}$, the $\delta D_{wax\ n\text{-alkanoic acid}}$ remained quite stable between 15-11 ka BP and after ~6 ka BP (Figure 4d). Both records agree with respect to the maximum at ~17 ka BP and the minimum between 10-7 ka BP (Figure 4d).

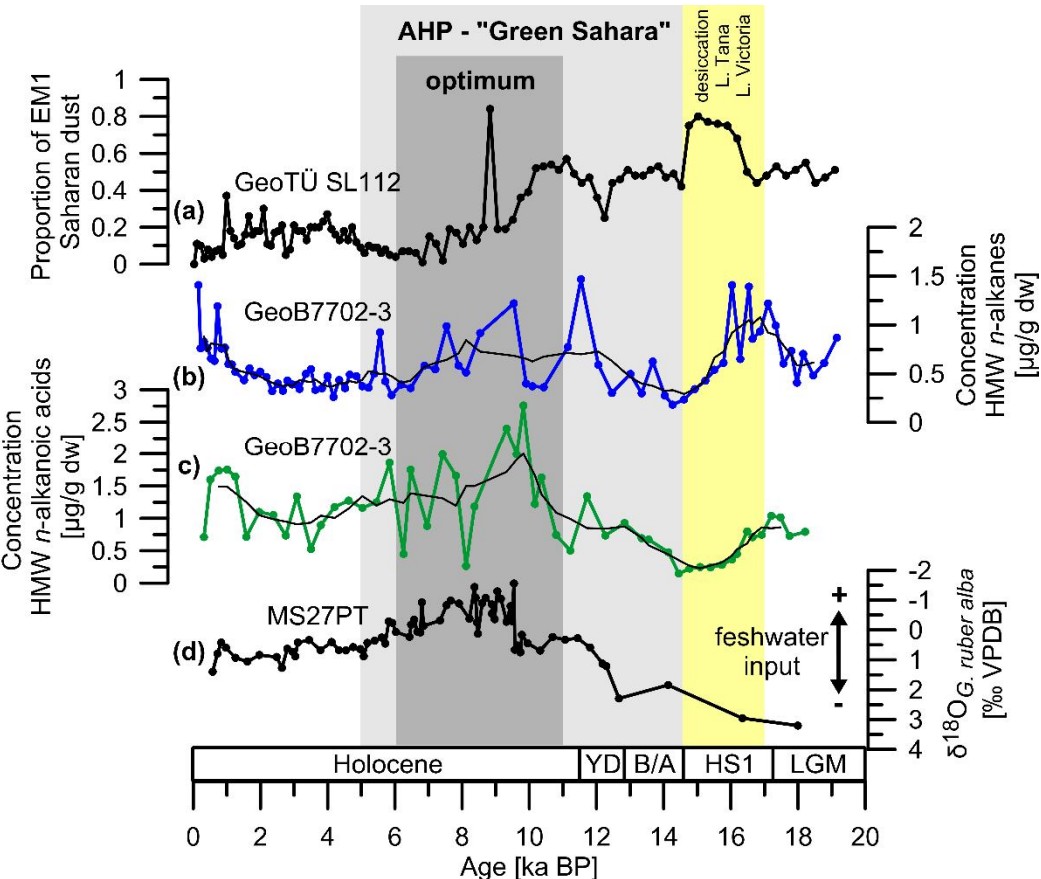

**Figure 3: Concentrations of HMW *n*-alkanoic acids (even numbered homologues $\Sigma C_{26:0}$-$C_{32:0}$; green (c)) and HMW *n*-alkanes (odd numbered homologues $\Sigma C_{29}$-$C_{33}$; blue (b)) in core GeoB7702-3. Black lines denote running averages. (a) endmember-modelling results from the grain size analysis in core GeoTü SL112 adopted from Hamann et al. (2008). The contribution of endmember 1 (EM1) to the silt fraction represents Saharan dust input to the Eastern Mediterranean (Hamann et al., 2008). (d) The oxygen isotopic composition of the planktic foraminifera species *Globigerinoides ruber alba* ($\delta^{18}O_{G.ruber\ alba}$) from core MS27PT (Figure 1c), Nile deep sea fan (Revel et al., 2010; 2015), is given to reflect freshwater runoff in the Nile River. YD: Younger Dryas stadial, B/A: Bølling/Allerød interstadial, HS1: Heinrich Stadial 1, LGM: Last Glacial Maximum. The light grey and dark grey shadings mark the episodes of the AHP and the Green Sahara, and their optimum. The yellow bar denotes the interval when Lakes Tana and Victoria desiccated (Stager et al., 2011; Lamb et al., 2007; Marshall et al., 2011; Williams et al., 2006).**

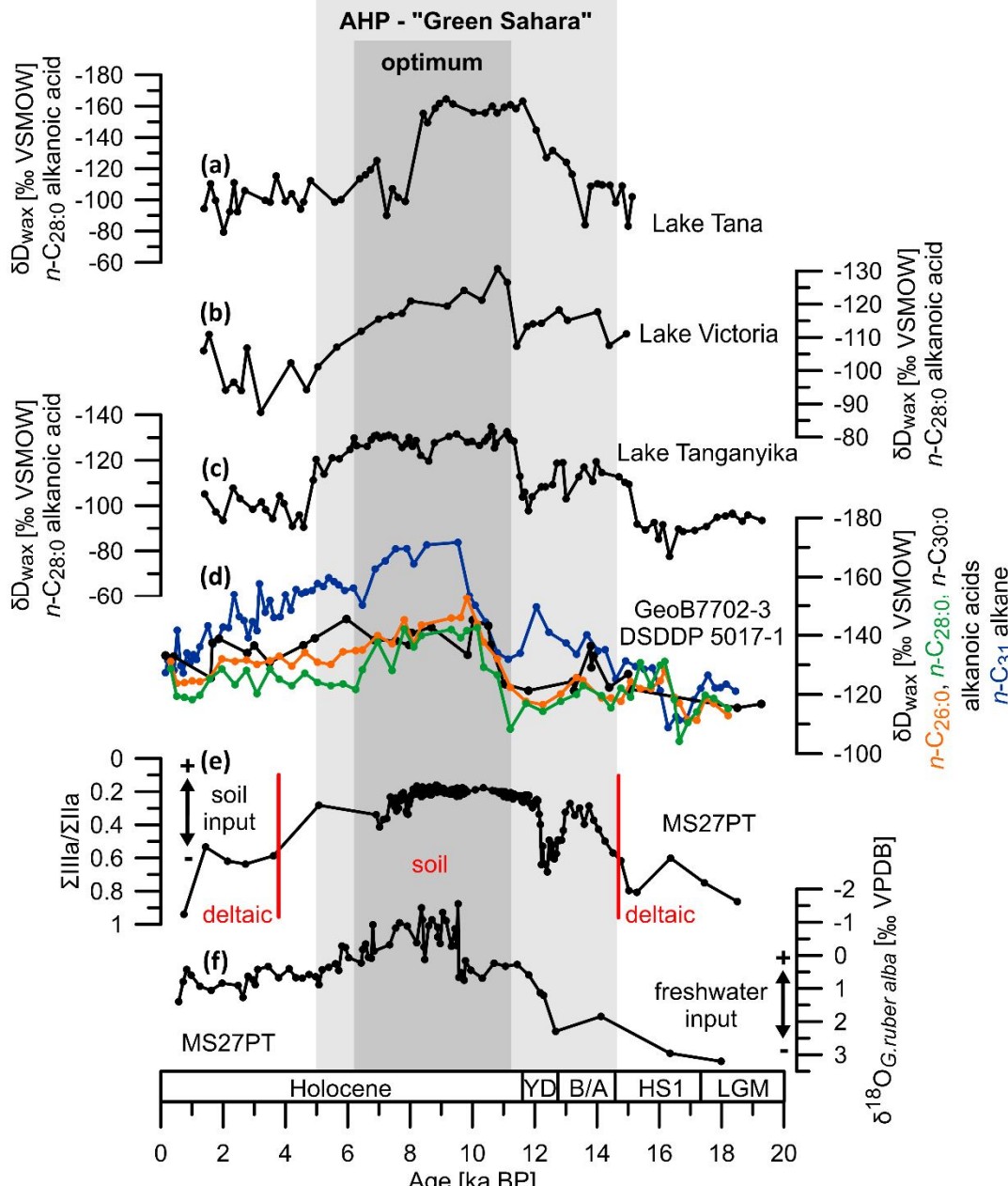


**Figure 4:** (a)-(c) δD$_{wax}$ records from Lakes Tana (Costa et al., 2014), Victoria (Berke et al., 2012) and Tanganyika (Tierney et al., 2008) reporting hydroclimate variability in tropical East Africa; (d) δD$_{wax}$ $_{n\text{-alkanoic acids}}$ (orange: $n$-C$_{26:0}$; green: $n$-C$_{28:0}$; this study) along with δD$_{wax\,n\text{-alkane}}$ (blue: $n$-C$_{31}$; Castañeda et al., 2016a; 2016b) from core GeoB7702-3 and δD$_{wax}$ based on the $n$-C30:0 alkanoic acid from core DSDDP 5017-1 from the Dead Sea (black; Tierney et al., 2022); (e) Ratio of tetra and penta-methylated brGDGTs from the Nile-deep sea fan (core MS27PT; Figure 1) reporting on the input of delta-derived organic matter versus material from the soils in the upper Nile-River catchment to the Eastern Mediterranean (Ménot et al., 2020); (f) oxygen isotopic composition of the planktic foraminifera species *Globigerinoides ruber alba* from the Levantine Basin (core MS27PT; Figure 1) used to reconstruct changes in salinity related to Nile River runoff (Revel et al., 2010; 2015). The light grey and dark grey shadings mark the episodes

of the AHP and the Green Sahara, and their optimum. **LGM:** Last Glacial Maximum; **HS1:** Heinrich Stadial 1; **B/A:** Bølling-
Allerød interstadial; **YD:** Younger Dryas stadial.

## 5. Discussion

$\delta D_{wax}$ is dependent to the $\delta D$ of the source water taken up by the plant during biosynthesis and thus can be used as tracer for
$\delta D$ of precipitation (e.g. Sachse et al., 2012 and references therein; Tierney et al., 2017). This, in turn, varies along with
hydrological processes including rainfall amount, evapotranspiration and moisture source but also temperature changes affect
the isotopic composition (Sachse et al., 2012). In low latitude regions $\delta D_{wax}$ is predominantly influenced by rainfall amount
(amount effect) as temperature effects are negligible (Sachse et al., 2012). This makes $\delta D_{wax}$ a powerful tool to reconstruct
glacial-interglacial fluctuations in rainfall amount and moisture source across Africa (Schefuß et al., 2005; Tierney et al., 2008;
Berke et al., 2012; Costa et al., 2014; Collins et al., 2013; Castañeda et al., 2016a). As for the arid Sahara, Tierney et al. (2017)
document that temperature effects play a subordinate role while precipitation exerts predominant control on the $\delta D_{wax}$.
However, changes in the relative abundance of $C_3$ versus $C_4$ plants may also overprint the hydrological signal in $\delta D_{wax}$ owing
to different fractionation factors in the Calvin and Hatch-Slack photosynthetic pathways (e.g. Sachse et al., 2012; Collins et
al., 2013). The $\delta^{13}C$ signature of leaf wax lipids is a common means to assess past changes in the relative contributions of $C_3$
versus $C_4$ plants in the catchment (Sachse et al., 2012; Collins et al., 2013) and to evaluate potential impacts of vegetation
changes on $\delta D_{wax}$ (e.g. Castañeda et al., 2016a; Collins et al, 2013). To explore potential effects of vegetation changes on our
$\delta D_{wax\ n\text{-alkanoic acid}}$ record, we correlate $\delta D_{wax\ n\text{-alkanoic acid}}$ and $\delta^{13}C_{wax\ n\text{-alkanoic acids}}$ (Figure S1). Considering that the linear correlation
yields low correlation coefficients ($R^2 < 0.5$; supplementary Figures S1 and S2), the impact of vegetation change on $\delta D_{wax\ n\text{-}}$
$_{alkanoic\ acid}$ is considered minor when the entire 18 ka are considered. Nevertheless, considerable impacts from vegetation changes
may have existed during shorter intervals despite the weak correlation. The amplitudes in the $\delta^{13}C_{wax\ n\text{-alkanoic acids}}$ are relatively
small (up to 3‰) rendering profound impacts of vegetation changes on $\delta D_{wax\ n\text{-alkanoic acid}}$ unlikely. When correcting $\delta D_{wax\ n\text{-}}$
$_{alkanoic\ acid}$ for vegetation changes (supplementary material) the patterns and amplitudes persist (Figures S1, S3). We conclude
that hydrologic variations exerted dominant control on $\delta D_{wax\ n\text{-alkanoic acid}}$ and that $\delta D_{wax\ n\text{-alkanoic acid}}$ is a robust proxy for changes
in rainfall and moisture source in the Nile River watershed throughout the past 18 ka.

To deduce deglacial changes in hydroclimate in the Nile-River catchment, we pair our $\delta D_{wax\ n\text{-alkanoic acid}}$ with the $\delta D_{wax\ n\text{-alkanes}}$
from Castañeda et al. (2016a). $\delta D_{wax\ n\text{-alkanes}}$ has been interpreted to reflect rainfall variability in the Nile-River watershed
associated with changes in the African summer monsoon (Castañeda et al., 2016a). The offset in $\delta D_{wax\ n\text{-alkanoic acid}}$ and $\delta D_{wax\ n\text{-}}$
$_{alkanes}$ between 15-1 ka BP (Figure 4d) suggests that the compounds recorded different hydrological developments, which in
turn implies that HMW *n*-alkanoic acids and HMW *n*-alkanes derive from different source areas. This conclusion seems
reasonable considering that the Nile-River watershed extends over several climate zones and covers strong hydrological
contrasts (tropical to subtropical and hyperarid desert). These zones are influenced by different atmospheric circulation patterns
with the westerlies influencing the middle and lower catchments and the African monsoon controlling precipitation in the
headwaters. Hence, the paired application of $\delta D_{wax\ n\text{-alkanoic acid}}$ and $\delta D_{wax\ n\text{-alkanes}}$ has the potential to reconstruct monsoonal along

with westerly precipitation in the Nile River watershed. It is crucial to evaluate from which section of the watershed HMW *n*-alkanoic acids and HMW *n*-alkanes derive to understand the hydrologic signals recorded in $\delta D_{\text{wax } n\text{-alkanoic acid}}$ and $\delta D_{\text{wax } n\text{-alkanes}}$.

### 5.1 Source areas of leaf-wax lipids in the Nile-River watershed

Leaf wax lipids reach the ocean mainly via fluvial discharge but also by aeolian transport (Schlünz and Schneider, 2000; Kusch et al., 2010; Schreuder et al., 2018). The latter is most important in open ocean settings where river discharge has little influence (Kusch et al., 2010; Hedges et al., 1997). Off large river systems, leaf-waxes are predominantly introduced by river discharge (e.g. Kusch et al., 2010) and accordingly leaf-waxes in the sediments of the Nile deep sea fan (e.g. Blanchet et al., 2014) and at site GeoB7702-2 (Castañeda et al., 2016a) have been interpreted as recorders of environmental change in the Nile River

watershed. However, in sediments of the southeastern Levantine basin the proportion of Saharan dust rises with increasing distance to the Nile delta (e.g. Krom et al., 1999). In core GeoB7702-3, concentrations of HMW *n*-alkanoic acids resembles the $\delta^{18}O$ record of the planktonic foraminifera species *Globigerinoides ruber alba* ($\delta^{18}O_{G. \text{ ruber alba}}$; Figure 3c,d) from a sediment core from the Nile deep sea fan (MS27PT, Figure 1) which tracks freshwater discharge from the Nile River (Revel et al., 2010). Concentrations and river runoff are relatively low during the LGM and the early deglaciation and increase between 14-10 ka BP. During the middle Holocene both decline (Figure 3c,d). Similarly, HMW *n*-alkane concentrations rise between 14-10 ka

BP and drop afterwards. This confirms that fluvial discharge was mainly responsible for the delivery of HMW *n*-alkanoic acids and HMW *n*-alkanes to the eastern Mediterranean. However, the abundances of HMW *n*-alkanes also have similarities with the proportion of Saharan dust in sediments of the southeastern Levantine Basin, in particular during the early deglaciation, before ~14 ka BP (Figure 3a, b). Thus, dust associated HMW *n*-alkanes were additionally deposited at site GeoB7702-3 at

these times. Considerable Saharan dust contributions are documented until 10 ka BP (Hamann et al., 2008; Figure 3a). Accordingly, aeolian transport may have been relevant to HMW *n*-alkanes from the last glacial until the early Holocene, possible dust sources to the southeastern Levantine Basin are the eastern Lybian Desert, central Algeria, Egypt and Sudan (Hamann et al., 2008). These regions are scarcely vegetated due to the vast extension of hyperarid desert. Therefore, dust associated HMW *n*-alkanes from these distal regions probably constitute a minor part in the HMW *n*-alkane pool at site

GeoB7702-3. Thus, environmental change in the Nile river watershed predominantly accounts for the signal recorded by the HMW *n*-alkanoic acids and HMW *n*-alkanes.

Leaf-wax based records from off river mouths often are interpreted as catchment integrating signals (e.g. Blanchet et al., 2014; Häggi et al., 2016; Hemmingway et al., 2016), as also done by Castañeda et al. (2016a) with respect to $\delta D_{\text{wax } n\text{-alkanes}}$ in core

GeoB7702-3. However, several studies indicate that this does not generally hold true for all types of lipids and not for all river systems (Hemmingway et al., 2016; Agrawal et al., 2014; Galy et al., 2011). It has been reported that leaf-wax lipids may be representative only for specific parts of a river catchment and source regions may vary between compound-classes and even between homologues within a compound class (e.g. Hemmingway et al., 2016; Agrawal et al., 2014). Hemmingway et al. (2016) found that HMW *n*-alkanoic acids originate from a local source in the Congo River watershed while HMW *n*-alkanes

serve as catchment-wide integrator. The differences between $\delta D_{wax\ n\text{-alkanoic acid}}$ and $\delta D_{wax\ n\text{-alkanes}}$ found in GeoB7702-3 strongly imply that the two lipid types derive from different source areas, at least for times when records of $\delta D_{wax\ n\text{-alkanoic acids}}$ and $\delta D_{wax\ n\text{-alkanes}}$ diverge (i.e. between 15-1 ka BP; Figure 4d). Similar $\delta D_{wax\ n\text{-alkanoic acid}}$ and $\delta D_{wax\ n\text{-alkanes}}$ are found before 15 ka BP and from 1 ka BP onwards (Figure 4d) suggesting that the compounds derive from the same area at these times. Since dust input from the Sahara was relatively stable between 14.6-10 ka BP (Figure 3a; Hamann et al., 2008), it is unlikely that aeolian

contributions from distal source regions outside the Nile River watershed account for the divergence in the $\delta D_{wax}$ records. Instead, the offset most likely stems from different source areas within the Nile River watershed. To identify the source areas of HMW $n$-alkanes and HMW $n$-alkanoic acids in GeoB7702-3 at present, we convert the $\delta D_{wax}$ values into $\delta D$ values of precipitation corrected for changes in vegetation and ice volume ($\delta D_{p\text{-vc-ic}}$) according to the methods described in the supplementary material.


We compare the near core-top values (here 15cm translating into 0.31 ka BP) to mean weighted $\delta D$ values of precipitation of the growing season ($\delta D_{p\text{-gs}}$) in the Nile catchment (Bowen et al., 2005). Bowen et al. (2005) define the growing season as months with mean temperatures >0°C. According to this definition, the values reported for the Nile River watershed are monthly weighted annual means since mean temperatures never drop below 0°C (https://en.climate-data.org). $\delta D_{p\text{-gs}}$ values

from different parts of the catchment and the results from our $\delta D_{p\text{-vc-ic}}$ are listed in Table 2. The results downcore are given in Figure S3. For the core top $\delta D_{p\text{-vc-ic}}$ is around -14 ‰ for the $n\text{-}C_{31}$ alkane and is -11 ‰ and -8 ‰ for the $n\text{-}C_{26:0}$ and $n\text{-}C_{28:0}$ alkanoic acids, respectively. This matches the isotopic composition of the Nile Delta where the predicted $\delta D_{p\text{-gs}}$ range between -15 and -11 ‰ (at Cairo, 30°3´N and Alexandria, 31°13´N; Bowen et al., 2005). Today, the Nile-Delta region receives the most deuterium-depleted precipitation in the entire watershed (Table 2; Bowen et al., 2005). In the middle section of the

catchment, along the main Nile (30°N-15°N), $\delta D_{p\text{-gs}}$ becomes progressively more enriched towards the South (-4 ‰ at Assuan, 24°6´N and 10 ‰ in Khartoum, 15°35´N). In the headwater region, precipitation is more depleted compared to the Main-Nile but still is enriched by 10-17 ‰ (3.5 ‰ Lake Tana, 12°N and -5.6 ‰ Lake Victoria, 1°S; Table 2) relative to $\delta D_{p\text{-gs}}$ in the delta (Bowen et al., 2005). As such, we infer that HMW $n$-alkanoic acids and HMW $n$-alkanes predominantly derive from the delta region at present.



**Table 2: Results for $\delta D_{p\text{-vc-ic}}$ calculated based on $\delta D_{wax}$ for GeoB7702-3 at 15 cm depth (~0.3 ka BP) together with $\delta D_{p\text{-gs}}$ values across the Nile River watershed. Locations listed are indicated in Figure 1.**

| Location | $\delta D_{p\text{-vc-ic}}$ [‰] | $\delta D_{p\text{-gs}}$[1] [‰] |
|---|---|---|
| GeoB7702-3; 15cm; $C_{31}$ *n*-alkane | -14.87 ± 4.95 | |
| GeoB7702-3; 15cm; $C_{26:0}$ *n*-alkanoic acid | -11.10 ± 3.14 | |
| GeoB7702-3; 15cm; $C_{28:0}$ *n*-alkanoic acid | -8.78 ± 2.51 | |
| Cairo/Nile Delta | | -14.6 to -11.4 |
| Alexandria/Nile Delta | | -13.1 |
| Assuan/Main Nile | | -4.4 |
| Dunqula/Main Nile | | 4.5 to 5.3 |
| Khartoum/Main Nile | | 9.9 |
| Kassala/Atbara | | 13.4 to 9.7 |
| Bahir Dar/Lake Tana | | 3.5 |
| Juba/White Nile | | 5.9 |
| Kampala /Lake Victoria | | -5.6 |

[1]: adopted from Bowen et al. (2005).

The offset between the $\delta D_{wax\ n\text{-alkanoic acid}}$ and $\delta D_{wax\ n\text{-alkanes}}$ between 15-1 ka BP (Figure 5e) suggests that either the source of the HMW *n*-alkanoic acid or the HMW *n*-alkanes changed during this timeframe. Considering that northeastern African hydroclimate saw drastic changes during the AHP (Shanahan et al., 2015; Berke et al., 2012; Tierney et al., 2008; Costa et al., 2014), it is very likely that the gradients in $\delta D_{p\text{-gs}}$ changed too. As such, the present-day distribution of $\delta D_{p\text{-gs}}$ cannot be used for paleo source-apportionment. Unfortunately, the detailed spatial distribution of past $\delta D_{p\text{-gs}}$ in the Nile River watershed remains unresolved, preventing us from using $\delta D_{p\text{-vc-ic}}$ to identify the source areas of the leaf-waxes during the last glacial and the deglaciation. Therefore, we compare our $\delta D_{wax}$ records to available $\delta D_{wax}$ data from Lakes Tana, Victoria and Tanganyika (Figure 1) and to a $\delta D_{wax}$ record from the Dead Sea representing hydroclimate variability in the headwaters, equatorial Africa and the Eastern Mediterranean realm (Figure 3a-d; Berke, et al., 2012; Costa et al., 2014; Tierney et al., 2008; 2022).

Our $\delta D_{wax\ n\text{-alkanoic acid}}$ has a remarkable resemblance with the $\delta D_{wax}$ record from the Dead Sea (Tierney et al., 2022; Figures 1, 4d). Plotted on the same scale the two records track each other sharing trends and amplitudes (Figure 4d). Even the range of values covered by the two records is almost the same varying between ca. -105 to -145 ‰ during the past 18 ka. These similarities suggest that the $\delta D_{wax\ n\text{-alkanoic acid}}$ in core GeoB7702-3 was determined by Mediterranean hydroclimate throughout the past 18 ka. This also implies that the HMW *n*-alkanoic acids were constantly sourced from the lower Nile River catchment. Due to the larger amplitude in the long-term development of the $\delta D_{wax\ n\text{-alkanes}}$ the range of values covered (-105 to -170 ‰) is larger than in the $\delta D_{wax\ n\text{-alkanoic acid}}$ (ca. -105 to -145 ‰). If the Dead-Sea $\delta D_{wax}$ and our $\delta D_{wax\ n\text{-alkanoic acid}}$ reflect the composition of the Mediterranean moisture (Tierney et al., 2022), the $\delta D_{wax\ n\text{-alkanes}}$ must have been influenced by more depleted rainfall, and thus by another moisture source, given the offsets between the Dead Sea record and the $\delta D_{wax\ n\text{-alkanes}}$ in core GeoB7702-3 (Figure 4d). At Lake Tana $\delta D_{wax}$ values of up to -160 ‰ have been recorded during the AHP providing evidence that in the

headwaters of the Blue Nile the rainfall was more depleted during the AHP than the Mediterranean precipitation (Figure 4a). It is most likely, that the HMW $n$-alkanes recorded climate variability in the southern reaches of the river where the summer

monsoon determines the hydroclimate. In equatorial Africa, decreasing $\delta D_{wax}$ in sediments from Lake Tanganyika (Tierney et al., 2008) point to increasing humidity from 15 to 11.5 ka BP (Figure 4c). Likewise, the levels from Lakes Tana and Victoria – which desiccated during the late glacial (Stager et al., 2011; Lamb et al., 2007) – rose at that time in response to a wetter climate (Williams et al., 2006; Marshall et al., 2011). The progressive decrease in $\delta D_{wax \, n\text{-alkanes}}$ starting at 15 ka BP thus is in harmony with the hydroclimate development in the headwaters during the early AHP further corroborating our inference that

the $\delta D_{wax \, n\text{-alkanes}}$ was substantially influenced by rainfall in the southern Nile River catchment. Accordingly, we propose that the HMW $n$-alkanes in core GeoB7702-3 received considerable contributions from the headwater region between ~15-1 ka BP, the time when $\delta D_{wax \, n\text{-alkanoic acid}}$ and $\delta D_{wax \, n\text{-alkanes}}$ deviate from each other. Before and afterwards both compounds derived predominantly from the delta. While the source of the HMW $n$-alkanes was dynamic during the past 18 ka, the source of the HMW $n$-alkanoic acids most likely was rather stable. Considerable contributions of HMW $n$-alkanoic acids from the headwater

region are unlikely. However, acknowledging that the Sahara was vegetated between ~11-5 ka BP (Kuper and Kröpelin, 2006; Watrin et al., 2009; Hély et al., 2014; Hamdan et al., 2016), the source region of the HMW $n$-alkanoic acids may have extended further south into the northern Sahara at this time. Likewise, the HMW $n$-alkanes probably received additional material from the middle catchment of the river. During the AHP, site GeoB7702-3 may also have received additional load from the Sinai Peninsula through Wadi El Arish. This drainage basin is rather dry at present but activated during the AHP (AbuBakr et al.,

2013; Muhs et al., 2013).

Our scenario regarding dynamics in HMW $n$-alkane provenance fits to findings from Ménot et al. (2020) who analyzed branched Glycerol Dialkyl Glycerol Tetraethers (brGDGTs) in the Nile deep sea fan to reconstruct soil input from the Nile River into the EM over the deglaciation. BrGDGTs are synthesized by bacteria thriving in peat and soils (Weijers et al., 2006, Martin et al., 2019) but can also be produced in-situ in rivers and estuarine settings and in the marine sediments (De Jonge et

al., 2014). Using the ratio between tetra- and penta- methylated brGDGTs ($\Sigma$IIIa/$\Sigma$IIa-ratio) Ménot et al. (2020) inferred that the compounds derived from the Nile delta between 20-14.5 ka BP and after 4 ka BP (Figure 3e). The interval of soil-derived input (14.5-4 ka BP; Ménot et al., 2020) matches the interval of diverging $\delta D_{wax \, n\text{-alkanoic acid}}$ and $\delta D_{wax \, n\text{-alkanes}}$ (Figure 4d) when considerable amounts of HMW $n$-alkanes from the upper and middle catchment were deposited in the EM sediments.

Being hydrophobic the HMW $n$-alkanes and HMW $n$-alkanoic acids are among the recalcitrant fraction of organic matter that

is preserved for long times in the geological record and consequently is able to survive riverine transport and intermediate storage in reservoirs prior to burial in the marine sediments (e.g. Eglinton and Eglinton, 2008). However, our north-south allocation regarding the sources of HMW $n$-alkanoic acids and HMW $n$-alkanoic acids in GeoB7702-3 suggests that in the Nile-River watershed, the most of the HMW $n$-alkanoic acids sourced from the upper catchment did not reach site GeoB7702-3, most likely due to degradation during riverine transport. This is in accordance with findings from Hemmingway et al. (2016)

who inferred that in the Congo-River catchment HMW $n$-alkanoic acids in suspended sediments from the outflow derive from local sources while HMW $n$-alkanes have a local and distal origin providing a more catchment integrated signal. Similarly,

Agrawal et al. (2014) suggested that in the Ganga-Bahamaputra River system HMW *n*-alkanoic acids from the Himalayan headwaters degrade during transport and get replaced by HMW *n*-alkanoic acids from the local floodplains. There is consensus that HMW *n*-alkanoic acids are more prone towards degradation than HMW *n*-alkanes (Meyers and Ishiwatari, 1993; Hoefs et al., 2002; Sinninghe-Damsté et al., 2002; Galy and Eglinton, 2011; Hemmingway et al., 2016) which probably accounts for the discrepancies in provenance of HMW *n*-alkanoic acids and HMW *n*-alkanes observed in these large river systems (e.g. Hemmingway et al., 2016) and at our study site.

## 5.2 Environmental drivers of HMW *n*-alkane provenance

There is compelling evidence that during the LGM and the early deglaciation, in particular Heinrich Stadial 1 (HS1), northeastern African climate was very arid (Stager et al., 2011; Castañeda et al., 2016; Tierney et al., 2008; Tierney and deMenocal, 2013; Revel et al., 2014; Ménot et al., 2020). Lakes Tana and Victoria, the sources of the Blue and White Nile tributaries, desiccated at 17-16 ka BP (Stager et al., 2011; Lamb et al., 2007) leading to a drastic reduction in runoff in the Nile River system relative to present (Williams et al., 2009; 2015) as documented by higher $\delta^{18}O_{G.\ ruber\ alba}$ (Figure 4f) offshore the Nile River mouth (Revel et al., 2010; 2015). At the same time deposition of Saharan dust peaked in the EM (Hamann et al., 2008). The weak fluvial activity in the Nile River likely restricted the source of the leaf-wax lipids to the Nile delta and led to minimal export of HMW *n*-alkanoic acids in eastern Mediterranean sediments (Figure 3c). Relatively high concentrations of HMW *n*-alkanes in core GeoB7702-3 despite the diminished fluvial activity (Figure 3b) can be best explained by aeolian transport of HMW *n*-alkanes sourced from the vegetated Nile delta region and floodplain deposits of the lower Nile River catchment considering that the input of Saharan dust to the eastern Mediterranean peaked at these times (Hamann et al., 2008; Figure 3a). $\delta D_{wax}$ records from Lake Tanganyika (Figure 3c) document that the East African climate became wetter around 14.5-15 ka BP (Tierney et al., 2008). In response to the wetter conditions the overflow of Lakes Tana and Victoria resumed around 15.5 and 14.5 ka BP (Williams et al., 2006; Marshall et al., 2011) and freshwater input from the Nile River increased accordingly as documented by lower $\delta^{18}O_{G.\ ruber\ alba}$ (Revel et al., 2010; 2014; 2015). The climate amelioration in northeastern Africa coincides with the onset of divergence between $\delta D_{wax\ n\text{-}alkanes}$ and $\delta D_{wax\ n\text{-}alkanoic\ acid}$ as well as with the switch from deltaic to soil-derived brGDGTs (Figure 4d,e). Concentrations of HMW *n*-alkanes increase along with decreasing $\delta^{18}O_{G.\ ruber\ alba}$ (Figure 3c,d) suggesting that enhanced fluvial energy in the headwaters probably increased erosion and export of organic matter in the upper reaches of the river as well as in the middle catchment where vegetation expanded into the Sahara in the course of the AHP. As mentioned above, a considerable amount of HMW *n*-alkanoic acids mobilized in the headwaters probably were degraded during riverine transport and did not reach the core site. Increasing concentrations of HMW *n*-alkanoic acids (Figure 3c) thus reflect intensified fluvial erosion in the delta and floodplains of the lower Nile catchment. During the late Holocene, at around 5-4 ka BP, drier conditions re-established in the headwaters and tropical East Africa (Tierney et al.,

2008, Costa et al., 2014, Berke et al., 2012) and Nile runoff reduced accordingly (Revel et al., 2010; 2015). Again, the delta became the predominant source of HMW $n$-alkanes and brGDGTs.

## 5.3 Hydroclimate development during the past 20 ka

For the reconstruction of the hydroclimate variability our findings imply that the HMW $n$-alkanoic acids and the HMW $n$-alkanes can be used to reconstruct the hydroclimatic developments in the lower and upper reaches of the Nile River in parallel (between ~14.5-4 ka BP). As elaborated earlier, the delta is predominantly influenced by Mediterranean winter precipitation. According to our inference that the HMW $n$-alkanoic acids consistently derive from the delta $\delta D_{wax\ n\text{-alkanoic acid}}$ should record changes in winter precipitation throughout the past 18 ka. By contrast the $\delta D_{wax\ n\text{-alkanes}}$ should predominantly be a summer
signal, as we infer that the HMW $n$-alkanes receive major contributions from the headwaters south of 15°N where the African summer monsoon controls precipitation. The application of paired $\delta D_{wax\ n\text{-alkanes}}$ and $\delta D_{wax\ n\text{-alkanoic acid}}$ allows to investigate how westerly (winter) and monsoonal (summer) precipitation evolved around the AHP.

## 5.4 early deglaciation – Heinrich Stadial 1 (18-14.6 ka BP)

The last glacial maximum (~21 ka BP; Clark et al., 2009) was a relatively moist interval in the southeastern Mediterranean
realm due to a more positive water balance (precipitation – evaporation) compared to today as indicated by higher levels of glacial Lake Lisan (paleo Dead Sea; Figure 1; Stein, 2001) and some pollen data from Dead Sea sediments (Langgut et al., 2021). In the northeastern Sahara, tufa deposits from the Eastern Desert provide evidence for sufficiently moist conditions to recharge aquifers (Hamdan and Brook, 2015). However, rainfall amount is debated (Ludwig and Hochmann, 2022; Kageyama et al., 2021; Goldsmith et al., 2017; Enzel et al., 2008). In case of reduced or similar rainfall amount, reduced evaporation must
have balanced reductions in annual rainfall (Ludwig and Hochmann, 2022). It is assumed that cold glacial conditions likely resulted in reduced evaporation and higher effective moisture during the LGM (Ludwig and Hochmann, 2021; Stockhecke et al., 2016). Moreover, the polar front and the westerlies were displaced southward relative to today (situated at ca. 40°N; Wang et al., 2018) owing to the presence of the Fennoscandian ice sheet (Wang et al., 2018; Kageyama et al, 2021). Accordingly, storms frequently steered moist air from the Atlantic Ocean and Mediterranean Sea to the southeastern Mediterranean
borderlands accounting for higher effective moisture and potentially also enhanced rainfall (Hamdan and Brook, 2015; Wang et al., 2018; Goldsmith et al., 2017; Kageyama et al., 2021; Enzel et al., 2008).
Northeastern African climate became harsher during the early deglaciation (from 19 ka onwards) with the most severe aridity occurring during HS1 (~18-14.5 ka BP; Tierney et al., 2008; Stager et al., 2008; Hamdan and Brook, 2015). Lakes Victoria and Tana desiccated (Stager et al., 2011; Lamb et al., 2007) and in the eastern Sahara, tufa formation ceased (Hamdan and
Brook, 2015). Also, in the Nile River delta conditions may have rapidly become more arid at the beginning of HS1 given the abrupt increase of $\delta D_{wax\ n\text{-alkanoic acid}}$ as well as the $\delta D_{wax\ n\text{-alkanes}}$ (Castañeda et al., 2016) at 17.8 ka BP (Figures 4d, 5d). Maximal values of $\delta D_{wax\ n\text{-alkanoic acid}}$ and $\delta D_{wax\ n\text{-alkanes}}$ suggest that also the Nile River delta experienced the most arid conditions since the LGM at that time. However, the drought was restricted to the first half of HS1. At ~16 ka BP, the climate rapidly

transitioned into a wet phase that lasted until the end of HS1 (14.6 ka BP) (Figures 4d, 5d). $\delta^{18}O$ of speleothems from the Soreq
Cave/Israel suggest a progressive increase in precipitation at the same time (Figure 5b). $\delta D_{wax\ n\text{-alkanoic acid}}$ values suggest hydroclimatic conditions similar to the late Holocene and today considering the similar values in $\delta D$ (-130 to -125‰). The arid episode attests to a decrease in precipitation and/or a decrease in evaporation. In turn, the reversal to more humid conditions was either caused by increased rainfall and/or reduced evaporation. The dry episode coincided with a moderate temperature rise as documented by $TEX_{86}$-based temperature reconstructions from core GeoB7702-3 (Castañeda et al., 2010a; Figure 5e). By contrast, the wet period occurred along with a cold phase in the eastern Mediterranean (Figure 5e) suggesting that early warming may have reduced moisture availability in the region by enhancing evaporation. Later, the reversal to colder conditions may have reduced evapotranspiration allowing for more effective moisture.

Next to temperature-induced changes in evaporation, variations in rainfall amount due to altered atmospheric circulation patterns are potential drivers that need to be evaluated. As noted by Bartov et al. (2003) lower air temperatures that possibly came along with low SSTs in the EM during HS1 (Bar-Mathews et al., 1999) would have minimized the cyclonic activity over the EM. A reduction of the intensity and frequency of Cyprus Lows would have reduced the advection of moist air and rainfall in the eastern Mediterranean realm fostering aridity. However, enhanced aridity is at odds with the isotopic depletion of the leaf waxes documented in core GeoB7702-3. If the local cyclonic activity was weakened, either effects of reduced evaporation must have balanced the reduced advection of moisture from the Cyprus Lows or another moisture source must have provided the region with rainfall. Interestingly, Italian speleothems imply that the delivery of Atlantic moisture to the central Mediterranean intensified from 16 ka BP onwards (Columbu et al., 2022). Analogous to the Italian site a stronger influence of the Atlantic moisture may have caused more humid conditions in the southeastern Mediterranean. The succession of dry and wet episodes within HS1recorded at our study site fits the growing view that HS1 evolved in two phases, the first one lasting from ~18.2- 16.2 ka and the second lasting from 16.2-14.5 ka BP (e.g. Naughton et al., 2023 and references within). Two-phase patterns with alternating dry and wet intervals have also been recorded by many marine and terrestrial archives in the entire Mediterranean realm (Valsecchi et al., 2012; Naughton et al., 2023; Naughton et al., 2009; 2016; Pérez-Mejías et al., 2021; Fletcher and Sanchez-Goñi et al., 2008) and even in western North America where they have been called "big dry" and "big wet" intervals (Broecker et al., 2009). However, the Mediterranean records are inconsistent regarding the order of wet and dry episodes indicating a complex hydroclimate development in the Mediterranean borderlands throughout HS1. For example, the Sea of Marmara/northeastern Mediterranean experienced wet conditions during the first phase of HS1 followed by drought during the second phase (Valsecchi et al., 2012). The opposite pattern is evident in our data at site GeoB7702-3. These regional differences are probably related to spatial variations of the Atlantic-Mediterranean storm track. Proxy-based and modeling studies suggest that the deglacial Mediterranean climate responded to abrupt climate fluctuations in the North Atlantic via atmospheric teleconnections involving the position of the westerly jet and the associated storm track (Columbu et al., 2022; Valsecchi et al., 2012; Li et al., 2019). According to Li et al. (2019) a weakening of the Atlantic Meridional Overturning Circulation (AMOC) during HS1 would push the westerlies northward. A northward movement of the westerlies and the Atlantic- storm track during the first half of HS1 could explain why the climate became drier in the Nile River delta

but concurrently wetter in the northeastern Mediterranean. A subsequent return to the South at ~16 ka BP would have reversed the situation with wet conditions in the Nile-River delta and concurrent aridity in the NE Mediterranean realm. Our inference

540 agrees with findings from the western Mediterranean basin since Naughton et al. (2023) conclude that opposing successions of dry and wet phases on the Iberian Peninsula during HS1 document a northward followed by a southward movement of the polar front and the associated westerly jet.

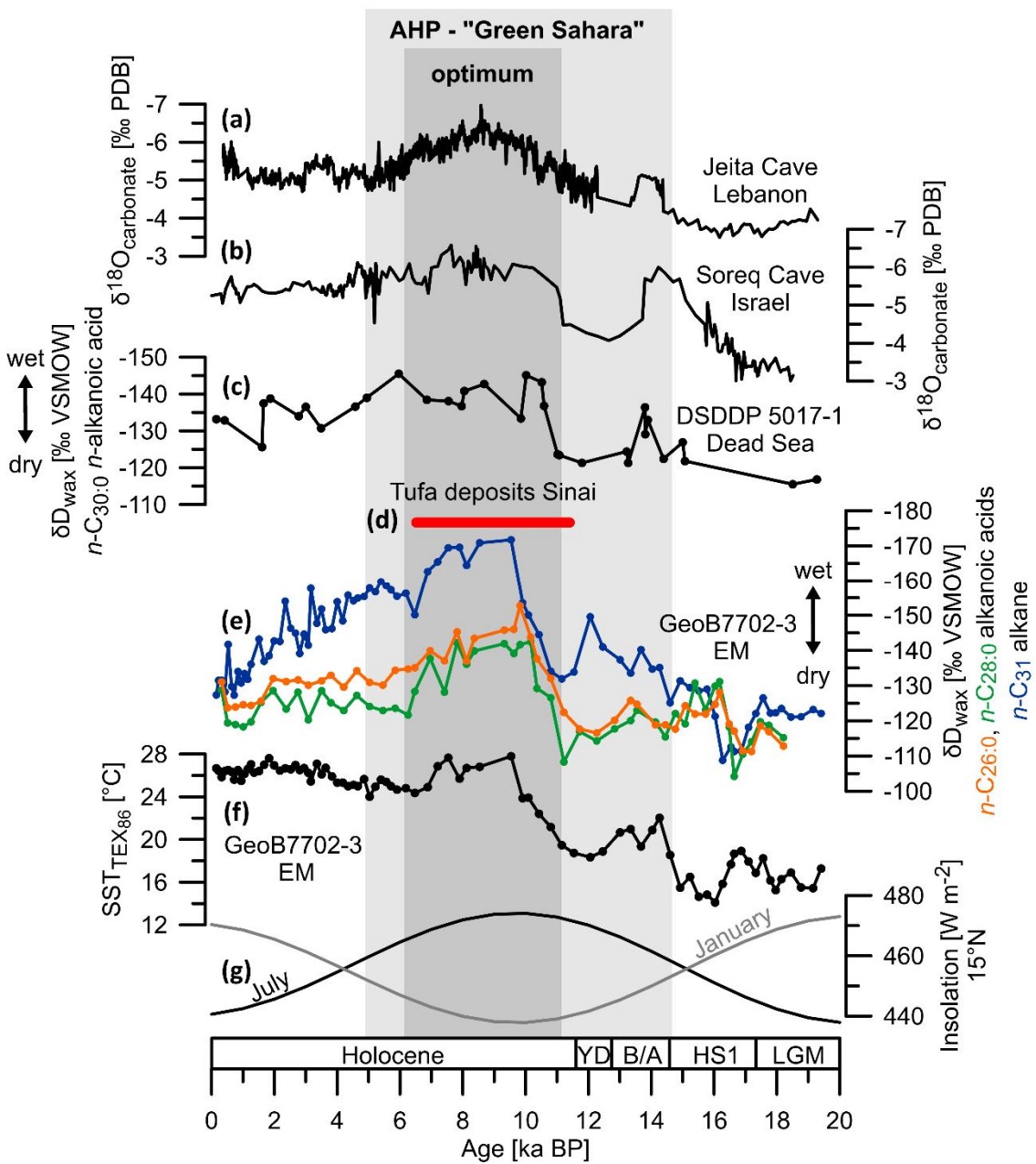

Figure 3: (a) Oxygen isotope records from speleothems ($\delta^{18}O_{carbonate}$) in Jeita Cave, Lebanon (Cheng et al., 2015); (b) Oxygen isotope records from speleothems in Soreq Cave, Israel (Bar-Mathews et al., 2003); (c) $\delta D_{wax}$ record based on the *n*-$C_{30:0}$ alkanoic acid in core DSDDP 5017-1 from the Dead Sea (Figure 1; Tierney et al., 2022) (d) The red bar marks the interval during which spring-tufa deposits formed on the Sinai Peninsula (Hamdan and Brook, 2015); (e) $\delta D_{wax}$ *n-*alkanoic acids (orange: *n*-$C_{26:0}$; green: *n*-$C_{28:0}$; this study) along with $\delta D_{wax}$ *n-*alkane (dark green: *n*-$C_{31}$; Castañeda et al., 2016a; 2016b) from core GeoB7702-3; (f) sea surface temperature (SST) in the Eastern Mediterranean (EM) based on $TEX_{86}$ from core GeoB7702-3 (Castañeda et al., 2010a; 2010c); (g) Insolation during January and July at 15°N from Berger and Loutre (1991). The light grey and dark grey shadings indicate the episodes of the AHP and the Green Sahara and their optimum. LGM: Last Glacial Maximum; HS1: Heinrich Stadial 1; B/A: Bølling-Allerød interstadial; YD: Younger Dryas stadial.

At the end of HS1, relatively dry conditions established in the Nile-Delta region and prevailed until 11.5 ka BP as documented by increased $\delta D_{wax}$ *n-alkanoic acid* (Figure 5d). The storm frequency and westerly moisture supply must have remained weak in the region throughout this period. Climate models propose that the polar front and the westerly jet moved northwards once the Fennoscandian Ice Sheet retreated (Wang et al., 2018; Kageyama et al., 2021). This northward movement may have diminished westerly-moisture supply to the Nile delta after 15 ka BP. Additionally, warming after HS1 (Figure 5e) may have intensified evapotranspiration fostering aridity in the region.

## 5.5 African Humid Period (14.5-5 ka BP)

Whereas $\delta D_{wax}$ *n-alkanoic acid* indicate relatively constant dryness after HS1, $\delta D_{wax}$ *n-alkanes* document a progressive depletion of plant waxes. As previously discussed by Castañeda et al. (2016a), this progressive depletion starting at ~15 ka BP most likely stems from increasing rainfall amount in the headwaters of the Nile-River catchment. This development agrees with increasing humid conditions at Lake Tanganyika, Tana and Victoria and has been interpreted as intensification of the African summer monsoon (Castañeda et al., 2016, Berke et al., 2012; Tierney et al., 2008; Costa et al., 2014). This hydroclimate amelioration led into in the AHP which lasted from ~14.5-5 ka BP and culminated between ~11-6 ka BP (e.g. deMenocal et al., 2006a; Tierney et al., 2017; Dupont and Schefuß, 2018). More negative $\delta D_{wax}$ *n-alkanoic acid* indicate that also the Nile Delta received substantially more rainfall during the AHP optimum, i.e. between 11.0-6.0 ka BP (Figure 4d).

A case study conducted by Breitenbach et al., (2010) in the Bay of Bengal/Indian Ocean shows that intensified river discharge due to heavy summer-monsoon rainfall leads to surface water freshening in the Ocean which in turn lowers the δD-signature of precipitation derived from these surface waters. Likewise, the discharge of isotopically light water from the Nile River watershed varied along with monsoon intensity during the Late Quaternary (Almogi-Labin et al., 2009). Isotopic variations in the Mediterranean surface waters may explain parts of the $\delta^{18}O$-signals recorded in speleothems from caves in Israel and the Lebanon (Cheng et al., 2012) and $\delta D_{wax}$ from the Dead Sea (Tierney et al., 2022). Considering that the $\delta^{18}O_{G.\ ruber\ alba}$ records from the Mediterranean Sea (Emeis et al., 2000; Revel et al., 2010; 2015) show massive surface freshening during the AHP resulting from enhanced monsoon rains in the Nile-River watershed, this effect could have potentially accounted for depleted $\delta D_{wax}$ *n-alkanoic acid* in addition to or instead of the amount effect. However, given that the $\delta^{18}O_{G.\ ruber\ alba}$ (Revel et al., 2010; 2015) and $\delta D_{wax}$ *n-alkanoic acid* show different temporal trends (Figure 4d,f), freshening of the Mediterranean probably constitutes only

a minor part of the signal in $\delta D_{wax\ n\text{-alkanoic acid}}$. Therefore, the minimum in $\delta D_{wax\ n\text{-alkanoic acid}}$ most likely attests to drastically increased rainfall amount in the Nile River delta during the AHP.

Interestingly, Tierney et al. (2022) found that the glacial-interglacial hydroclimate is quite sensitive to temperature driven changes in the atmospheric evaporative demand. Warming enhances evapotranspiration and reduces effective moisture. TEX$_{86}$-based SST reconstructions at site GeoB7702-3 show pronounced warming at 11-10 ka BP (Figure 5e; Castañeda et al.,

2010a). Consequently, a more negative water balance due to elevated evapotranspiration would be expected for the Levant. However, an increase in evapotranspiration and reduction in effective moisture would cause an isotopic enrichment of the leaf waxes rather than a depletion (Sachse et al., 2012). As such, the decrease in $\delta D_{wax\ n\text{-alkanoic acid}}$ reported at site GeoB7702-3 clearly documents that increased rainfall must have exceeded any potential increases in evaporation thereby leading to a more positive water balance.

It is well constrained that the ITCZ and summer monsoon shifted northward during the AHP supplying the Sahara with seasonal rainfall (e.g. deMenocal et al., 2006b; Menviel et al., 2021; Braconnot et al., 2007; Blanchet et al., 2021; Tierney et al., 2017; Kuper and Kröpelin, 2006; Hamdan and Brook, 2015). However, the northernmost extent of the monsoon is debated. Today the northernmost limit is at 14°N. In many climate simulations for the AHP the monsoon reaches ~24°N (e.g. Pausata et al., 2016; Chandan and Peltier, 2020; Thompson et al., 2021; Hopcroft et al., 2017; Dallmeyer et al., 2020). This is a more modest

extent compared to pollen-based reconstructions which indicate the limit at 23-28°N (Bartlein et al., 2011; Hély et al., 2014). However, some climate models acknowledging several land-surface feedbacks set the northernmost boundary as far north as 30-31°N over northwestern Africa (Pausata et al., 2016). Likewise, analysing isotopic compositions of speleothems and leaf-wax lipids researchers concluded that the monsoon fringe expanded up to 27-31°N over northwestern Africa (Sha et al., 2019; Tierney et al., 2017), yet this interpretation recently was challenged by pollen data from northern Morocco (Cheddadi et al.,

2021). As for northeastern Africa climate models consistently compute very small positive anomalies or no change (e.g. Perez-Sanz et al., 2014; Dallmeyer et al., 2020; Pausata et al., 2016), even when wet conditions reach as far north as 30-31°N over the northwestern Sahara (Pausata et al., 2016). Records of Saharan dust input to the North Atlantic and the Red Sea also point to a greater extent of the monsoonal rains in the northwestern than in the northeastern Sahara (Palchan and Torfstein, 2019; McGee et al., 2013). The west-east gradient in humidity has been attributed to the tilted axis of the monsoonal rainbelt (Pausata

et al., 2016; Dallmeyer et al., 2020). From the climate model perspective monsoonal rains did not reach the Nile delta region during the AHP. This view is consistent with dust accumulation rates in the Red Sea which set the limit of monsoonal rains at 22°N over northeastern Africa (Palchan and Torfstein, 2019). Further support comes from Holocene spring tufa deposits in the Eastern Desert and the Sinai Peninsula (Hamdan and Brook, 2015). The tufas from the Eastern Desert (~25°N) were fed from monsoonal precipitation whereas tufa from the Sinai Peninsula (at ~27°N) precipitated from springs recharged by

Mediterranean rainfall (Hamdan and Brook, 2015). Moreover, salinity reconstructions from the Red Sea (Arz et al., 2003) suggest that the Mediterranean was the predominant moisture source to north easternmost Africa during the Holocene. Collectively, models and proxies suggest that the monsoonal rainbelt never reached the Nile Delta region. Thus, instead of

monsoonal rain, increased winter precipitation associated with the westerlies most likely was responsible for the minimum in $\delta D_{wax\ n\text{-alkanoic acid}}$ between 11-6.5 ka BP.


### 5.5.1 Controls on enhanced winter precipitation

The rapid switch to wetter conditions at 11 ka BP as well as the abrupt return to dry conditions at 6.5 ka BP displayed by the $\delta D_{wax\ n\text{-alkanoic acid}}$ occurred congruently with a warm phase in the EM between 10-7 ka BP documented by the $TEX_{86}$-temperature proxy from GeoB7702-3 (Castañeda et al., 2010a). The warm SST would have enhanced evaporation over the
Mediterranean Sea and at the same time would have led to a higher water-holding capacity of the atmosphere in virtue of the Claudius-Clapeyron relation (e.g. Trenberth et al., 2003; Lenderink and Meijgaard, 2008). Additionally, warming would have strengthened the formation of Cyprus Lows over the Levantine Basin. Both effects would have increased moisture delivery to the Nile-delta region. Records from the Middle East (Figure 4a,b), southern Europe and northwest Africa provide evidence that mild and wet conditions were widespread across the Mediterranean borderlands during the early Holocene (Cheddadi et
al., 2021, Wagner et al., 2019, Bar-Mathews et al., 2003; Cheng et al., 2015). Today, relatively mild and wet winters in the Mediterranean realm are associated with negative phases of the NAO-Index during which the westerlies weaken and the Atlantic storm track is situated south steering Atlantic moisture and storms frequently into the Mediterranean (Figure 1). As invoked by several data- as well as model-based studies, negative NAO-like circulation patterns probably prevailed during the early Holocene promoting intense winter precipitation in the Mediterranean borderlands (Kutzbach et al., 2014, Arz et al.,
2003, Dixit et al., 2020, Wassenburg et al., 2016). Previous work suggests that the southward displacement of the westerlies and the associated storm trajectory was a response to decreasing Northern Hemisphere winter insolation and thus a result of precessional forcing (Kutzbach et al., 2014; Wagner et al., 2019; Li et al., 2019). Analysing lacustrine sediments from Lake Ohrid (Balkan Peninsula) Wagner et al. (2019) proposed that the Mediterranean winter precipitation varied in-phase with the African summer monsoon strength over glacial-interglacial cycles as both were driven by precessional and thus insolation
forcing. The monsoon responded to rising summer insolation changes and Mediterranean precipitation to weakened winter insolation (Wagner et al., 2019; Cheng et al., 2016). Indeed, $\delta D_{wax\ n\text{-alkanoic acid}}$ and $\delta D_{wax\ n\text{-alkanes}}$ from core GeoB7702-3 confirm that Mediterranean winter rains and the African summer monsoon (Castañeda et al., 2016a; Costa et al., 2014, Berke et al., 2012, Tierney et al., 2008) were concurrently strong in the Nile River watershed during the AHP optimum but wet conditions began later (~11 ka BP) and also ended earlier (~6 ka BP) in the delta than in the upper river catchment (~14.5-5 ka BP; Figures
4a-d, 5d). The relatively short wet phase recorded in our $\delta D_{wax\ n\text{-alkanoic acid}}$ agrees with pronounced freshening of the northern Red Sea which has been interpreted to result from enhanced winter rainfall over Egypt and the Sinai Peninsula (Arz et al., 2003). Moreover, spring tufa deposits formed on the Sinai Peninsula only between 9.0-6.7 ka BP (Figure 5c) and were fed by Mediterranean moisture (Hamdan and Brook, 2015). Together these data show that in the northeastern Sahara wet conditions were restricted to the AHP optimum (11.0-6.5 ka BP). Likely, the storm track reached the northeastern Sahara only when the
difference between summer insolation and winter insolation was maximal (Figures 5a,b,c,d). Interestingly, speleothem $\delta^{18}O$-

records from caves in Lebanon (Jeita Cave; Figure 1) suggest that the hydroclimate may have ameliorated slightly earlier than in the Nile Delta, i.e. around 12 ka BP (Figure 5a,d) while Soreq Cave and $\delta D_{wax}$ from the Dead Sea indicate an increase of humid conditions at the same time as $\delta D_{wax\ n\text{-alkanoic acid}}$ in core GeoB7702-3 (Figure 5b,d). Likewise, wet conditions lasted longer at Jeita Cave than in the Nile-River delta region (Figure 5a, d). The staggered onset and termination of wet conditions in the south-eastern Mediterranean realm may attest to the progressive southward and subsequent northward migration of the storm track if the temporal offset does not stem from uncertainties in the chronologies.

### 5.5.2 Implications for the "Green Sahara" (11-5 ka BP)

Providing insights into the development of summer and winter precipitation in the Nile-River watershed, our findings are important for the understanding of the moisture sources which sustained the "Green Sahara". As elaborated above and by Castañeda et al. (2016a), the $\delta D_{wax\ n\text{-alkane}}$ attest to enhanced monsoon rainfall in the upper Nile-River watershed. By contrast, our $\delta D_{wax\ n\text{-alkane}}$ is a signal of winter rainfall from the lower catchment. It is widely accepted that the northward migration and intensification of the summer monsoon was critical to trigger plant growth in the Sahara during the AHP (Braconnot et al., 2007; Menviel et al., 2021; deMenocal et al., 2000b; Shanahan et al., 2015; Sha et al., 2019; Tierney et al., 2017). However, discrepancies between proxy data and climate simulations regarding precipitation and vegetation anomalies (Chandan et al., 2020; Braconnot et al., 2007; Hopcroft et al., 2017; Perez-Sanz et al., 2014; Cheddadi et al., 2021; Hély et al., 2014) question that the monsoon was the only moisture source to sustain a green Sahara (Cheddadi et al., 2021). According to climate model simulations monsoonal precipitation may not have provided enough moisture to sustain vegetation beyond ~24°N given mismatches with proxy data (Chandan and Peltier, 2020; Braconnot et al., 2007; Perez-Sanz et al., 2014; Cheddadi et al., 2021). Savannah centred between 20-25°N partly reaching to ~28-31°N (Larrasoaña et al., 2013; Hély et al,. 2014; Giraudi et al., 2013; Hamdan et al., 2016). The northeastern Sahara, south of the Nile Delta, turned into a semi-arid landscape covered by patchy vegetation comprising desert and open grass savannah as well as Mediterranean vegetation near the coast (Hamdan et al., 2016; Larrasoaña et al., 2013). Wetlands and trees likely flourished near permanent water bodies such as the Faiyum Lake (Hamdan et al., 2016; Watrin et al., 2009). Thus, since the monsoon extended no further than ~22°N over northeastern Africa (e.g. Palchan and Torfstein, 2019; Arz et al., 2003), it likely was insufficient to support such vegetation in the lower Nile-River Basin (between 22-31°N).

Our $\delta D_{wax\ n\text{-alkanoic acid}}$ show that Mediterranean winter rainfall enhanced during the AHP suggesting that enhanced winter rainfall created favourable conditions for the establishment of the desert-steppe vegetation around the Nile Delta. The modern winter rainfall zone is restricted to a small area along the Mediterranean coast (between ~30-31°N over northeastern Africa). If winter rains not only intensified but also protruded deeper into the Sahara they may have provided the moisture needed to sustain the grasslands north of 22°N in the northeastern Sahara (Cheddadi et al., 2021). Seeking evidence that the winter rainfall zone extended deeper into the Sahara, the information could be potentially discovered in the $\delta D_{wax\ n\text{-alkane}}$ considering that HMW n-alkanes receive substantial contributions from the upper and middle catchment while the HMW n-alkanoic acids derive mainly from the delta region. Between 11-10 ka BP – at the onset of the AHP and green Sahara optimum – the $\delta D_{wax\ n\text{-}}$

alkane shows a reversal to enriched δD values (Figure 5d). The reversal to enriched $\delta D_{\text{wax } n\text{-alkane}}$ may either be interpreted as a reduction in rainfall amount or as a change in the moisture source. A similar reversal is not recorded in the $\delta D_{\text{wax}}$ records from Lakes Tana, Victoria and Tanganyika (Berke et al., 2012, Tierney et al., 2008, Costa et al., 2014). These records even suggest that the monsoonal rainfall reached its maximum at this time. Therefore, it is unlikely that the reversal in $\delta D_{\text{wax } n\text{-alkane}}$ between 11.5-10.5 ka BP stemmed from a weakened monsoon. Lake Tanganyika as well as other East African archives show that the monsoon weakened during the Younger Dryas (13.8-11.5 kaBP; Figure 4b,c; Tierney et al., 2008; Berke et al., 2014; Talbot et al., 2007). Analogously, Castañeda et al. (2016a) suggested the reversal would attest to drier conditions during Younger-Dryas stadial. However, we rule out that the enrichment in $\delta D_{\text{wax } n\text{-alkane}}$ is a delayed climate deterioration associated with the Younger Dryas considering our revised age model for core GeoB7702-3 (uncertainties: ~ ±370 yrs at 279 cm, meadian age: 11,457 yrs, Table 1). Given the absence of a comparable signal in the $\delta D_{\text{wax}}$ records from the headwaters, the Dead Sea and the $\delta D_{\text{wax } n\text{-alkanoic acid}}$ (Figure 5a-d), the signal most likely was generated in the middle part of the Nile catchment, i.e. in the Sahara, south of the Nile delta. The reversal coincides with the beginning decrease of $\delta D_{\text{wax } n\text{-alkanoic acid}}$ and consequently with the starting intensification of winter precipitation at 11 ka BP (Figure 5d). As such, the reversal to more enriched $\delta D_{\text{wax } n\text{-alkane}}$ between 11.5-10.5 ka BP probably attests to increased influence of Mediterranean precipitation in the Sahara.

The general enrichment of $\delta D_{\text{wax } n\text{-alkanoic acid}}$ relative to $\delta D_{\text{wax } n\text{-alkane}}$ during the early AHP (~14.5-11 ka BP) provides evidence that the Mediterranean moisture was isotopically heavier than moisture from the Indian and Atlantic Oceans, the latter feeding the African summer monsoonal rainfall (Figure 1). This further argues for Mediterranean influence on the $\delta D_{\text{wax } n\text{-alkane}}$ during this period. The minimum in $\delta D_{\text{wax } n\text{-alkane}}$ succeeding the reversal (10.5-7 ka BP; Figure 5d) then probably also stemmed from the amplification of winter precipitation, given the strong similarity to the minimum in $\delta D_{\text{wax } n\text{-alkanoic acid}}$ that occurred concurrently (Figures 4d, 5d). This inference is supported by the absence of similar minima in $\delta D_{\text{wax}}$ from Lakes Tana, Victoria and Tanganyika at these times (Figure 4a-c; Berke et al., 2012, Costa et al., 2014, Tierney et al., 2008). Seeking to explain the minimum in $\delta D_{\text{wax } n\text{-alkane}}$ between 10 and 7 ka BP Castañeda et al. (2016a) invoked an eastward shift of the Congo Air Boundary during the AHP optimum, which would enhance the influence of Atlantic moisture over the headwaters. However, the striking similarity between the $\delta D_{\text{wax } n\text{-alkane}}$ and $\delta D_{\text{wax } n\text{-alkanoic acid}}$ is a profound argument for the influence of winter precipitation. Acknowledging that $\delta D_{\text{wax } n\text{-alkane}}$ remain depleted relative to the $\delta D_{\text{wax } n\text{-alkanoic acid}}$ despite the influence of Mediterranean moisture (Figure 4d, 5d), monsoon precipitation probably still substantially influenced the $\delta D_{\text{wax } n\text{-alkanes-signal}}$. We conclude that the $\delta D_{\text{wax } n\text{-alkanes}}$ recorded monsoonal precipitation superimposed by Mediterranean winter precipitation between ~12-6 ka BP, broadly consistent with the AHP optimum. As such, we deduce that this episode corresponds to the timeframe of the southward extension of Mediterranean winter rainfall zone.

Our finding supports earlier studies proposing southward extended winter rains during the early-to-mid Holocene in northeast Africa. South of the Nile Delta, the formation of the paleo Faiyum-Lake (Egypt, Figure 1) which is dated to ~10 ka BP (Hamdan et al., 2016; Hassan et al., 2012) hints to increased winter rains south of the present-day limit of the Mediterranean rainfall zone (i.e. approximately the southern tip of the Nile delta, Cairo). However, the inundation of the Faiyum depression to a large extent is a result from high Nile floods which are linked to monsoonal rainfall in the headwaters and local precipitation is

considered of minor importance for the lake formation (Hamdan et al., 2016; Wendorf and Schild, 1976; Hassan, 1986). Lacustrine sediments at the northern shore of the modern Lake Quarun, the relict of the paleo Faiyum Lake, attest to Wadi activity in the region which is independent from Nile flow and thus indicative of stronger Mediterranean winter rainfall (Hamdan et al., 2020; Koopmann et al., 2016). Moreover, several authors argue that the arrival of East Asian domesticated cereals in the Faiyum region during the early Holocene was enabled by a southward extension of the winter rainfall zone as most of these crops rely on precipitation in winter to grow and on long daylight hours in summer (Phillipps et al., 2016; Shirai et al., 2016). Salinity reconstructions from the Red Sea (Art et al., 2003) and tufa formation on the southern Sinai Peninsula (Hamdan and Brook, 2015) provide evidence that Mediterranean winter rains reached further south than nowadays. Cheddadi et al. (2021) proposed that the Mediterranean winter rainfall may have reached as far south as 18-24°N based on a combination of proxy data from Morocco and vegetation modelling. Accordingly, winter rainfall may have played a crucial role in sustaining the green Sahara. Our data confirm that during the AHP optimum an interplay of monsoon (summer) and Mediterranean (winter) rainfall existed in the Nile River watershed which may have provided the northeastern Sahara with sufficient rainfall throughout the year allowing for plants to occupy the nowadays barren desert.

## 6. Conclusions

We analysed $\delta D_{wax\ n\text{-alkanoic acid}}$ in core GeoB7702-3 from the Eastern Mediterranean and generated a new continuous record of winter precipitation in the Nile-River delta for the past 18 ka BP, in a region where continuous records are sparse. By comparison to previously published records of $\delta D_{wax\ n\text{-alkanes}}$ from the same sediment core, we gain new information about the provenance of leaf-wax lipids in the Nile River watershed and about the interplay of Mediterranean (winter) and monsoonal (summer) precipitation around the AHP and their significance for the genesis and sustaining of the Green Sahara. Our key findings can be summarized as follows:

1) HMW $n$-alkanoic acids predominantly derived from the delta during the past 18 ka while the source of the HMW $n$-alkanes varied through time as a function of river runoff and vegetation coverage in the Sahara. Between 15-4 ka BP the HMW $n$-alkanes received major contributions from the headwaters and the Sahara once the river runoff was relatively high due to intensified summer rains. Before and after this interval HMW $n$-alkanes derived from the delta region, like the HMW $n$-alkanoic acids. Around the AHP (14.5-5 ka BP) the paired application of $\delta D_{wax\ n\text{-alkanoic acids}}$ and $\delta D_{wax\ n\text{-alkane}}$ allows to reconstruct winter rainfall ($\delta D_{wax\ n\text{-alkanoic acid}}$) along with summer monsoonal precipitation ($\delta D_{wax\ n\text{-alkane}}$) in the Nile-River watershed

2) HS1 occurred in two phases in the Nile River delta due to rapid variations in the position of the westerly jet and the associated storm track over the Mediterranean. First the climate became arid due to a northward shift of the storm track. At 16 ka BP wet conditions established as the storm track was pushed south. Afterwards the climate remained relatively dry until 11 ka BP as the westerly jet moved north in response to ice sheet retreat.

3) In the Nile-delta we find evidence for increased Mediterranean winter rainfall between 11-6 ka BP corresponding the optimum of the AHP. Amplified winter precipitation resulted from a combination of regional and large-scale factors. Local

cyclogenesis (Cyprus Lows) promoted by elevated SSTs and southward displaced westerlies (negative NAO like pattern) steering Atlantic storms into the Mediterranean intensified the moisture delivery and precipitation in the Nile-River delta region. The episode of intensified rainfall was much shorter in the delta region than in the southern Nile catchment, where the summer monsoon determined precipitation. In the headwaters the hydroclimate had already begun to ameliorate as early as 750  14.5 ka BP. Likely, winter precipitation only increased when the storm track reached its southernmost position during maximal difference between NH summer and NH winter insolation.

4) At the time of Sahara greening and enhanced precipitation in the delta region (11-6 ka BP) the monsoonal signal in the $\delta D_{wax \, n\text{-alkane}}$ was superimposed by intensified Mediterranean precipitation in the Sahara. Given that we find proof for concurrently intensified winter and summer precipitation in the Nile-River watershed and infer that westerly precipitation was 755 enhanced not only in the delta region but also penetrated farther south into the Sahara, our data support the recently raised hypothesis that monsoonal precipitation only together with winter precipitation could allow for vegetation to occupy the Sahara Desert.

**Data availability**

The data generated in this study will be accessible on the PANGAEA database:
760 https://doi.pangaea.de/10.1594/PANGAEA.961489

**Competing interests**

The authors declare that they have no conflict of interest.

**Author contributions**

765 ES designed the study together with VM, GM and JP. JP provided sediment material for core GeoB7702-3. VM carried out the sample preparation and analysis in the laboratories and conducted the data processing. IC and SS analyzed the concentrations of HMW *n*-alkanes. VM drafted the manuscript with contributions from all co-authors.

**Acknowledgements**

770 We thank the captain and the crew of RV Meteor for their effort during cruise M52/2. Pushpak Nadar is thanked for his valuable help during processing the biomarker and foraminifera samples in the laboratories at MARUM. Ralph Kreutz is acknowledged for his support during the isotope analysis. We thank Liz Bonk, Hendrik Grotheer and Torben Gentz for their efforts during the AMS measurements. This work was funded by the DFG Research Center/Excellence Cluster "The Ocean Floor – Earth's Uncharted Interface". We are grateful to two anonymous reviewers and the Editor Christo Buizert for 775 constructive comments during the review process which helped to improve the quality of the manuscript.

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
