# Peer review of "Evolution of winter precipitation in the Nile-River watershed since the last glacial"

_Climate of the Past, 2023_

## Author Response (AR1)

Reply to the Editor

Dear Christo,

Thank you very much for your comments on our manuscript and for inviting us to submit a revised version. We replied to all reviewer comments and your comment. In the following we describe how we addressed your concerns. The lines refer to the track-changes version of the revised document. Our answers are given in italics.

Kind regards

Vera Meyer and co-authors

Dear Authors,

Your manuscript has now been seen by two reviewers. Both evaluations were overall quite positive, but identify some issues that need to be addressed. Please respond to all of the reviewer comments, as well as my additional minor comments below this note. I will likely be encouraging you to submit a revised version of your manuscript, so feel free to respond to the reviewer comments in the form of proposed changes to the manuscript.

I look forward to reading your response.

All the best, Christo Buizert (CP editor)

Additional comments from the editor:

While the d2H is a good proxy for low-latitude monsoon precipitation via the amount effect, in the mid-latitudes water isotope fractionation is dominated by the temperature effect. This may also be true for the westerlies that you invoke for the winter precipitation. Can you please elaborate on what you think is driving the isotope signature of the winter precipitation?

*As described in section S2 (supplementary material) we retained from performing a temperature correction. It is correct that temperature exerts control on the dD of the lipids and that this control increases with increasing latitude. Considering that Tierney et al. (2017) show that the amount effect dominates over temperature effects in the arid Sahara the temperature effect accordingly should have minimal influence on the dD in our record. Moreover, finding a suitable temperature record to perform corrections in the Nile region is difficult because temperature estimates provide an inconsistent picture regarding the amplitude of glacial-Holocene warming and some of the records do not cover the last 18 ka (Lakes Tana and Victoria). There are TEX$_{86}$-based and alkenone-based sea surface temperature (SST) reconstructions from the eastern Mediterranean Sea and the Red Sea but their amplitude of change varies between 6-10°C (Castaneda et al., 2010; Arz et al., 2003; Mathews et al., 2021) which is up to 2 times as much as climate models suggest for the change in mean annual surface temperature (5-6°C; Kageyama et al., 2021). On top of that, the SST-proxies are probably also affected by seasonal and other biases that may lead to an overestimation of the true amplitude of deglacial warming (Castañeda et al., 2010).*

*Therefore, we might overcorrect by using these SST reconstructions from the Mediterranean to correct the δDwax for the temperature effect. A brGDGT-based record on mean annual air temperature was established on core GeoB7702-3 (Castañeda et al., 2016) but brGDGTs were analyzed with an outdated HPLC-method that does not allow for adequate separation of the brGDGT compounds. The record may thus be significantly influenced by in situ production and may not reflect MAT-changes in the Nile-River watershed. Given these circumstances we decided not to correct for temperature. Since lower temperature conditions during the glacial lead to isotopically depleted rainfall the reconstructed dDp of the last glacial can be regarded as minimum values, respectively maximal rainfall estimates.*

*We added a few lines to section S2 (supplementary material).*

Figure 1A: Could it be that the NAO+ and NAO- labels have been switched? I believe the description in the text (section 5.5.1) is correct. Please double check.

*Yes, you are right. Thank you for pointing out. In the figure + and – were accidentally mixed up while the NAO states are correctly described in the text. We changed Figure 1.*

Line 13: I prefer to use d2H rather than dD, as it reduces unnecessary jargon and is more consistent with notation of all other isotope ratios. Please give both notations at this first use of dD. It would be too much to ask you to change this throughout the manuscript, but please keep this in mind for future publications.

*As you suggested, we stick to dD in the new version but we add d2H when dD is introduced in the introduction (line 113).*

Line 185: "set to 50": what are the units on this number?

*In BACON the "acc.mean" prior is given in yrs/cm. We added this to the value (line 185).*

Line 218: I assume you apply this to correct for thee mean d2H of ocean water? The LR04 stack gives the d18O of benthic foraminifera, which is impacted by both the d18O of the ocean, as well as the temperature effect. The latter effect needs to be removed – how did you do this? I assume you multiply by 8 to get the d2H magnitude?

*We performed the ice volume correction according to Ruan et al., 2019. We assume the larger ice volume introduced an enrichment of 1‰ (Schrag et al., 1996) to the glacial ocean. The glacial-Holocene amplitude of the benthic stack is 1.8‰. This is normalized to 1‰ and accordingly this step corrects for any other effect on the benthic d18O, including bottom-water temperature. Multiplying this by 8 (global meteoric water line) yields a change of 8‰ in dD due to ice-volume changes. In order to avoid any questions regarding the temperature correction, we added a citation to line 219.*

Line 346: Source areas (remove plural "s")

*Done.*

Line 577: Note that a similar division of HS1 is seen in North America; Broecker referred to these as the "big wet" and "big dry" (Broecker et al., 2009)

*Thank you for mentioning this study. That is very interesting and we were not aware of these findings before. We included a reference to Broecker et al. (2009) in the discussion about HS1 (lines 579).*

References:

Arz, H., Lamy, F., Pätzold, J., Müller, P. and Prins, M. A.: Mediterranean Moisture Source for an Early-Holocene Humid Period in the Northern Res Sea, Science, 300, 118–121, 2003.

Broecker, W. S., McGee, D., Adams, K. D., Cheng, H., Edwards, R. L., Oviatt, C. G., & Quade, J. (2009). A Great Basin-wide dry episode during the first half of the Mystery Interval? *Quaternary Science Reviews, 28*(25-26),

Castañeda, I. S., Schefuß, E., Pätzold, J., Sinninghe Damsté, J. S., Weldeab, S. and Schouten, S.: Millennial-scale sea surface temperature changes in the eastern Mediterranean (Nile River Delta region) over the last 27,000 years, Paleoceanography, 25(1), 1–13, doi:10.1029/2009PA001740, 2010.

Castañeda, I. S., Schouten, S., Pätzold, J., Lucassen, F., Kasemann, S., Kuhlmann, H. and Schefuß, E.: Hydroclimate variability in the Nile River Basin during the past 28,000 years, Earth Planet. Sci. Lett., 438, 47–56, doi:10.1016/j.epsl.2015.12.014, 2016.

Kageyama, M., Harrison, S. P., Kapsch, M. L., Lofverstrom, M., Lora, J. M., Mikolajewicz, U., Sherriff-Tadano, S., Vadsaria, T., Abe-Ouchi, A., Bouttes, N., Chandan, D., Gregoire, L. J., Ivanovic, R. F., Izumi, K., Legrande, A. N., Lhardy, F., Lohmann, G., Morozova, P. A., Ohgaito, R., Paul, A., Richard Peltier, W., Poulsen, C. J., Quiquet, A., Roche, D. M., Shi, X., Tierney, J. E., Valdes, P. J., Volodin, E. and Zhu, J.: The PMIP4 Last Glacial Maximum experiments: Preliminary results and comparison with the PMIP3 simulations, Clim. Past, 17(3), 1065–1089, doi:10.5194/cp-17-1065-2021, 2021.

Matthews, A., Affek, H., Ayalon, A., Vonhof, H., Bar-Matthews, M., 2021. Eastern Mediterranean climate change deduced from the Soreq Cave fluid inclusion stable isotopes and carbonate clumped isotopes record of the last 160 ka. Quat. Sci. Rev. 272, 107223

Ruan, Y., Mohtadi, M., van der Kaas, S., Dupont, L. M., Hebbeln, D. and Schefuß, E.: Differential hydro-climatic evolution of East Javanese ecosystems over the past 22,000 years, Quat. Sci. Rev., 218, 49-60, doi: 10.1016/j.quascirev.2019.06.015, 2019.

Schrag, D. P., Hampt, G. and Murray, D. W.: Pore fluid constraints on the temperature and oxygen isotopic composition of the glacial ocean, Science, 272, 1930–1932, 1996.

Tierney, J. E., Pausata, F. S. R. and De Menocal, P. B.: Rainfall regimes of the Green Sahara, Sci. Adv., 3(1), 1–10, doi:10.1126/sciadv.1601503, 2017. (supplementary material)

Reviewer 1

Dear reviewer,
thank you very much for your constructive suggestions how to improve our manuscript. In the following we explain how we address your concerns in the revised version of the article. Our reply is written in italics. Line numbers refer to the track-changes version of the manuscript.

Kind regards,

Vera Meyer and Co-authors

This paper presents new analyses of the C26 and C28 alkanoic acids from a core the Levantine basin (GeoB7702-3). There was a previously published C-31 alkane record from this site, so analysis of these new chain lengths offers an interesting complement. More could be done with analysis of the offset between different chair lengths, though they largely reflect a very similar signal. There is a need to hone explanation of the mechanisms underlying changes in the record, but otherwise it is a very nice study.

*In the new version of our manuscript we expanded our discussion about the source apportionment (section 5.1) and on the mechanisms controlling the evolution of winter precipitation in the Nile River delta (section 5.4-5.5.2).*

The introduction is very clear and very well written. However, I would try to discuss more uncertainties about winter rainfall in the last glacial and mid-Holocene. See the dynamical papers linked below.

*Thank you for pointing to these studies. As requested, we added a short paragraph about the uncertainties regarding the balance between evaporation and precipitation during the LGM to the introduction (lines 80-92). We also extend the discussion as the reviewer asked for honing the discussion about the mechanisms controlling the hydroclimate in Nile delta region. We shed more light on the P-E balance by adding few lines focusing on the effects of evaporation on dD, an aspect which did not receive much attention in the previous version. We also cite the studies in this context (lines 528-541; 558-564; 648-655).*

Methods:

Table 1: I would actually rather see the age model figure (e.g. the pdf output from BACON) and a table like this in the Supplementary information, especially since many of the dates are already published in other sources).

*As requested, we included a figure depicting the BACON output (line 241). However, we keep Table 1 in the main article to present all information about the age model at the same spot.*

3.2 Lipid extraction: do you worry about an aquatic signature with C26 and C28 at all? There is some literature suggesting that mid-chains alkanes could be produced by submerged macrophytes (see Ficken et al 2000)?

*You are correct that mid-chain homologues of n-alkanes have been found to sometimes stem from aquatic origin. Biases from algal and bacterial origin are also possible for mid-chain HMW n-alkanoic acids but predominantly affect homologues with a chain length <28 (e.g. Kusch et al., 2010). Biases on homologues with a greater chain length are quite unlikely (Kusch et al., 2010) which is why the $n\text{-}C_{28:0}$ is very often used for the reconstruction of terrigenous environments (e.g. Costa et al., 2014, Berke et al., 2012; Tierney et al., 2008). In our core the $n\text{-}C_{26:0}$ and $n\text{-}C_{28:0}$ have very similar $\delta D$ signals suggesting that they have the same source. The $\delta^{13}C$ values of the $n\text{-}C_{26:0}$ and $n\text{-}C_{28:0}$ vary between -22.6 and -26.9‰ values of submerged plants often are > -20‰ (e.g. Liu et al., 2022). Therefore, we consider both homologues reliable proxies for hydroclimate in the Nile-River watershed.*

Discussion:

I disagree with the logic that if d13C and dD are not significantly correlated then it is not necessary to consider the impact of vegetation change. A step change in d13C can possibly mask a change in dD of precipitation in a leaf wax record if there is a shift in the dominant physiological pathway during an interval when dD of leaf wax appears relatively complacent. I think a more defensible argument to make is that the amplitude of the carbon isotope change in your record is very small, and therefore a constant epsilon/apparent fractionation would be more appropriate at this site.

*You are right stating that the amplitude of change in the d13C records are very small and therefore do not introduce large uncertainties into the dDwax. This is best documented by the shape of the dDwax record when correcting it for vegetation and ice-volume changes. The trends and amplitudes are similar to the original dDwax record which shows that the effect of changing vegetation only marginally affected the dDwax signal. We added a short paragraph to lines 327-331 extending the discussion about the impact of vegetation changes.*

Line 422: I am not sure I agree that the alkanoic acids and alkanes really show completely dissimilar patterns - if you think they do, it would be good to describe where the dissimilarities arise. To my eye, it appears that the n-acids and n-alkanes show similar trends but different amplitudes of variability.

*"Dissimilar patterns" refers to the different developments during the deglaciation and the Holocene as described in section 4 (results). The amplitude of the long-term development is larger in the n-alkanes than in the fatty acids. In the revised version we re-wrote the discussion about the source apportionment (section 5.1) because we implemented the Dead Sea record from Tierney et al. (2022). In the course of re-writing lines 415-423 were deleted.*

While it is lower resolution, there is actually an n-acid leaf wax record of hydrogen isotopes from the Dead Sea that was recently published (Tierney et al 2022, Quaternary Science Reviews). It might be useful to compare your findings to that record, as it will show whether over time the isotopic signature at GeoB7702-3 more strongly reflects African records influenced by the AHP or Levantine records.

*Thank you very much for pointing to this study. The record is a very valuable support for our interpretations regarding Mediterranean winter precipitation in our records. It is very useful to have inferences on hydroclimate from the Eastern Mediterranean realm that are based on the same proxy. We included the dD record from the Dead Sea into Figures 4 and 5. As elaborated in the discussion (lines 433-454) it broadly tracks changes in our dDwax n-alkanoic acids records. The range of absolute values covered in the two records are in agreement. By contrast, the HMW n-alkanes exceed the range matching values found in the records of Lake Tana. This underscores our inference that HMW n-alkanoic acids reflect Mediterranean winter rainfall while HMW n-alkanes are substantially influenced by summer monsoon rain.*

Section 5.1 and 5.2:

The idea of using modern precipitation isotopes to essentially 'fingerprint' the source area from different leaf wax isotopes is an interesting one. I would like to see this information incorporated into the figures a little more - you could potentially plot dD of precipitation (inferred by applying a constant epsilon to your wax records) and plot modern values of dDp from different parts of the Nile catchment on this plot.

*When we developed the manuscript we also considered plotting the modern values on the axis of the dDp record. However, we discarded this idea since the source apportionment based on the end-members from the different parts of the watershed actually only works for the present-day situation. Therefore, we decided to present the data in a table. While at present the distribution of dDp along the watershed is known in detail, it is unknown for the deglacation and the LGM. However, we know that rainfall amount changed drastically as the monsoon intensified and changes in the end-members must be expected. Given these uncertainties on paleo end-members of dDp we would like to avoid to present the modern values along with the paleo dDp estimates from our core in order to avoid confusion. By including the Dead Sea dD record to our Figures 4 and 5 the source apportionment is strengthened (see our response to the first comment on section 5.1) and lines 433-454 in the manuscript. In the revised version we explain why we do not use the modern dDp values for the source apportionment through time (lines 423-433).*

Given the strong focus on organic matter source and provenance, it would also be nice to see changes in concentration of different leaf waxes over time plotted in one of the main text figures (at least for C26 and C28, the chain lengths measured in this study)

*For the revised version we created a new figure showing the concentrations of HMW n-alkanes and HMW n-alkanoic acids (Figure 3). We plotted the data together with records of Saharan dust input to the eastern Mediterranean and with d18O-records on planktic foraminifera representing Nile River runoff. Sections 4, 5.1 and 5.2 were extended by a few lines discussing concentrations of n-alkanes and n-alkanoic acids in context of fluvial and aeolian transport (lines 254-263, 352-368, 379-383, 492-469, 503-508). We also added information about the analytical methods used for the quantification of the lipids (lines 202-207, 222-235). While the concentrations of n-alkanoic acids were analyzed in our study, the concentrations of n-alkanes were measured by Isla Castañeda and Stefan Schouten several years ago but were unpublished. For providing their data, Isla and Stefan became co-authors of our manuscript.*

5.3:

Again, I think more analysis is needed to clearly show that alkanoic acids vs. alkanes are picking up different signatures of winter vs. summer precipitation. Perhaps I am missing something, but Table 2 seems to suggest that the modern near-foretop values of both the C-31 alkane and the C-26 and C-28 alkanoic acids appear similar to precipitation isotopes in the Delta compared to the isotopic signature farther south?

*Yes, that is correct and a key finding for the discussion about provenance. In section 5.1 we described that both types of compounds derive predominantly from the delta at present and also during the LGM when the dDwax and dDp match. During the deglaciation the offset of the dDwax and dDp between n-alkanes and n-alkanoic acids suggests that the source areas of the n-alkanes and n-alkanoic acids differed substantially. The comparison with the dD record from the Dead Sea provides further support for the inference that the HMW n-alkanoic acids record changes in the Mediterranean hydroclimate. The records show similar trends with parallel occurrence of the wet -phase at the time of the AHP. Also, the range of dD values covered by the two records is almost the same while the HMW n-alkanes and wax records from headwaters (in particular Lake Tana) have a larger range. By including references to the Dead Sea record we strengthen the discussion about the signals of winter versus summer precipitation in our dDwax. Lines 433-454. We also added the Dead Sea record to Figures 4 and 5.*

A broader point to consider: it appears you are primarily interpreting modern precipitation isotope seasonality in terms of seasonality (e.g. summer vs. winter end members) - how does this contrast with the 'amount effect' often used in paleoclimate studies?

*Based on the offsets of the dDwax of n-alkanes and n-alkanoic acids during the deglaciation we conclude that the compound types reflect climate change in two precipitation regimes, i.e. in the winter rainfall zone (Nile delta region; n-alkanoic acids) and the summer monsoon zone in the headwaters (n-alkanes) as described in the source apportionment in section 5.1.. The deglacial variability in the two records is a result of the amount effect in the two catchments, i.e. winter rainfall amount in the delta region (alkanoic acids) and monsoonal rainfall amount in the headwaters (n-alkanes). In short, the offset between the homologues stems from different source areas, the variability in the records reflects the amount effect.*

A few dynamical studies that may be useful to consider

Ludwig, P. and Hochman, A., 2022. Last glacial maximum hydro-climate and cyclone characteristics in the Levant: a regional modelling perspective. Environmental Research Letters, 17(1), p.014053.

Goldsmith, Y., Polissar, P.J., Ayalon, A., Bar-Matthews, M., DeMenocal, P.B. and Broecker, W.S., 2017. The modern and Last Glacial Maximum hydrological cycles of the Eastern Mediterranean and the Levant from a water isotope perspective. Earth and Planetary Science Letters, 457, pp.302-312.

Kusch, S., Rethemeyer, J., Schefuß, E. and Mollenhauer, G.: Controls on the age of vascular plant biomarkers in Black Sea sediments, Geochim. Cosmochim. Acta, 74(24), 7031–7047, doi:10.1016/j.gca.2010.09.005, 2010.

Liu, H., Liu, J., Hu, J., Cao, Y., Xiao, S. and Liu, W.: Systematical $\delta 13C$ investigations of TOC in aquatic plants, DIC and dissolved $CO2$ in lake water from three Tibetian Plateau lakes, Egol. Indic., 140, 109060, 2022.

Response letter to Reviewer 2

Dear Reviewer,

Thank you very much for your constructive suggestions how to improve the quality of the manuscript. You identified some paragraphs which were a bit to vaguely written. By adding a few more details to the discussion in order to improve the clarity. Below we describe how we address your concerns in the revised version. Our responses are given in italics. Line numbers refer to the track-changes version of the article.

Kind regards

Vera Meyer and co-authors

The authors use stable hydrogen isotopic compositions of high molecular weight n-alkanoic acids in a marine sediment core from the Eastern Mediterranean, to provide a continuous record for winter precipitation of the Nile watershed since the end of the last glacial period.

Overall the paper is well written and easy to follow; however, I second Reviewer #1 on explanations regarding the mechanisms: I feel the authors easily draw conclusions on large scale dynamical changes with little support, except for some sparse paleoclimate archives (e.g. storm track boundary located at 31-33°N based on just one proxy from Jeita cave – LL622-623). Given the authors confusion of ITCZ and WAM (see below), they should be very careful before making atmospheric or ocean dynamics claims. Regarding the analysis of the paleoclimate archives per se, it is outside my field of expertise so I leave it to the other reviewer(s).

*You are right that some single inferences in section 5 were based on few data and might have been a bit too vague. Therefore, we removed the paragraph encompassing lines 609-623. All remaining inferences made on large-scale atmospheric circulation patterns are based on several studies from the western, central and eastern Mediterranean realm as well as on some results from general circulation models (sections 5.4 and 5.5.1). As we discuss our data in context to those records and findings we feel that the conclusions made are more solidly justified than the conclusions the reviewer was criticizing in the previous version.*

*Reviewer #1 also asked for a more detailed discussion about the mechanisms controlling the hydroclimate in the Nile-Delta region. In order to acknowledge his concerns we included some paragraphs about the development of the P-E balance in the region (lines 528-541; 558-564; 648-655).*

Furthermore, I feel the way the authors cited paper is a bit random. I could not verify all of them but in the few cases I did, the references were not appropriate:

*Regarding your concerns on our way of citing, we went into more detail describing the findings of the studies cited. We feel that some passages were kept too short and produced confusion. By extending the discussion we hope to avoid misunderstandings. We rephrased*

*passages in the introduction (section 1) and section 5. Below, we elaborate how we addressed the instances specifically.*

- In L29 the authors cited de Menocal et al 2000 when referring to "dramatic oscillations between dry and wet climate states in the course of glacial-interglacial cycles". De Menocal et al. 2000 just showed one oscillation during the last deglatiation! The authors should rather cite Larassoana et al. 2013 or other more relevant paper.

*We additionally cite Menviel et al. (2021); Ziegler et al. (2010) and Larrasoaña et al. (2013), which all mention the last interglacial AHP.*

- In LL71-74 the authors wrote "However, the comparison of simulations and vegetation reconstructions shows that climate models probably underestimate precipitation in the northern Sahara and that the summer monsoon alone may have been insufficient to sustain a vegetated Sahara (…)" and refer among others to Hely et al. 2014 who did not claim something like that.

*In order to avoid any confusion about the references, we extended the paragraph and give more details about precipitation estimates from models and proxies (lines 50-70).*

- In LL653-654 the authors claim "Some studies suggest that the monsoon fringe even expanded up to 31°N (Sha et al., 2019; Tierney et al., 2017) which would mean that the Nile-River delta (situated at 30-31°N; Figure 1) became influenced by the African summer monsoon.". Both papers use paleoclimate archives from western Africa and several modelling studies have shown that eastern Africa remained much drier than the western part (e.g., Pausata et al., 2016; Dallmeyer et al., 2020*).

  *We re-wrote and extended the discussion about the northernmost extent of the WAM in this paragraph to clarify what models and data suggest regarding the northwestern and northeastern Sahara (lines 658-687).*

  Then in LL654-656 the authors wrote "However, in most climate simulations the northernmost position of the ITCZ is located at ~24°N (e.g. Pausata et al., 2016) which is also corroborated by proxy data (Hamdan and Brook, 2015; Cheddadi et al., 2021)." The authors are mixing up the ITCZ extension with the northern most location of the ITCZ (same in line 740 where they confuse the ITCZ for the WAM). The two sentences the authors wrote are not in contradiction: the very paper the authors cite for the ITCZ location at 24°N (Pausata et al., 2016) is the same showing the northern most extension of the WAM reaching up to 31°N, which is also cited in Tierney et al. 2017. Even with the WAM reaching up to 31°N in western Sahara, in the Nile Delta the summer rainfall in their model did not exceed 0.5 mm/day, i.e. less than 60 mm over the entire summer season.

*Thanks for pointing out these erroneous denotations. Of course, the northernmost position of the WAM was meant in the sentence in line 654-656. In the new version this is corrected (line 662). As indicated in our response to the previous comment, we re-wrote the paragraph in lines 658-687. In doing so (lines 658-687), we give details about different inferences regarding the northernmost extent of the WAM (having in mind the definition given in Pausata et al., 2016, i.e. the northernmost latitude that the monsoon reaches). In this paragraph we also mention the west-east gradient in humidity (lines 574-576) and conclude*

*that the WAM has never reached the Nile Delta region during the AHP (lines 684-688). Moreover, we rephrased section 5.5.6 as we intended to more appropriately cite other studies (see our response to the second comment). Line 740 was changed to: "According to climate model simulations monsoonal precipitation may not have provided enough moisture to sustain vegetation beyond ~24°N given mismatches with proxy data (Chandan and Peltier, 2020; Braconnot et al., 2007; Perez-Sanz et al., 2014; Cheddadi et al., 2021)" (lines 793-795).*

The authors should also cite Hely et al. 2014 here.

*We cite Hely in line 792.*

> Finally, to get a better understanding of the relation between WAM and ITCZ I strongly suggest the authors to read this review article: Geen, R., Bordoni, S., Battisti, D. S., & Hui, K. (2020). Monsoons, ITCZs, and the concept of the global monsoon. Reviews of Geophysics, 58, e2020RG000700. https://doi.org/10.1029/2020RG000700.

*Thank you for mentioning this informative review paper.*

L147 In Alexandria does not rain only 118 mm/year! It rains almost twice about 190-200 mm/year. I then checked climate.data reference provided by the authors where one can see an annual average of 181 mm/year, which is more reasonable. Please correct it.

*Thank you for pointing out! We corrected the typo.*

L152 Mai --> May.

*Corrected.*

L168 For clarity I would repeat in the sentence in which areas "Warm and wet winters are generally associated with negative NAO-states while cold and dry winters occur during positive phases."

*Done.*

In Figure 1 panel A the NAO+ should be NAO- and vice versa.

*You are right, the NAO states were accidentally mixed up during the figure production. Thanks for pointing out! However, in the text the attribution is correct. We corrected Figure 1.*

Figure 4: the Green Sahara is also known as African Humid Period. I do not understand while the authors are giving them two difference period ranges.

The reference to 11-5 ka BP for the green Sahara was based upon e.g. Watrin et al., 2009*. However, in the revised version we reconsidered the interval and now attribute the green Sahara to 14.6-5 ka BP. In Figure 4 and 5 we marked the optimum of the AHP and green Sahara with the dark grey shading. The dashed lines have been removed.

North Africa refers to specific countries so it's better to use northern Africa (the n is not capital). Moreover, the authors sometimes use capital letters (Northwest Africa in L77) sometime lowercase (northeast Africa in L78). I suggest using northwestern, northeastern, northern Africa to avoid confusion with the political definition of those regions.

*As recommended we use northwestern, northeastern, northern, southern, etc. throughout the manuscript.*

Westerly Jet, why jet is with a capital letter?

*Written in lower case letters, now.*

L691 the authors refer to a warm spell for the warming period in the EM between 10 – 7 years ago. However, warm spells are periods characterized by **several days** of very warm temperatures compared to local or regional averages. Warm spell is not the right term to use in a climate context.

*We replaced "spell" by "phase".*

* Dallmeyer, A., Claussen, M., Lorenz, S., and Shanahan, T.: The end of the African humid period as seen by transient comprehensive Earth system model simulation of the last 8000 years, Clim. Past, 16, 117–120, https://doi.org/10.5194/cp-16-117-2020, 2020.

Watrin, J., Lézine, A. M., Hély, C., Cour, P., Ballouche, A., Duzer, D., Elenga, H., Frédoux, A., Guinet, P., Jahns, S., Kadomura, H., Maley, J., Mercuri, A. M., Pons, I. A., Reynaud-Farrera, I., Ritchie, J. C., Salzmann, U., Schulz, E., Tossou, M. G., Vincens, A. and Waller, M. P.: Plant migration and plant communities at the time of the "green Sahara," Comptes Rendus - Geosci., 341(8–9), 656–670, doi:10.1016/j.crte.2009.06.007, 2009.

---

## Author Response (AR2)

Dear authors,

Your revised manuscript has now been seen by one of the original reviewers. The reviewer suggests that more changes are needed before the manuscript can be published. In particular, the reviewer suggests that the climate dynamics interpretation is misrepresented in the manuscript, and in some cases even internally contradictory. Please provide a revised version of your manuscript that incorporates the additional reviewer comments.

One of the suggestions is to look at transient climate model simulations of the last deglaciation (the TraCE and/or i-TraCE simulations). In particular the isotope-enabled i-TraCE experiment could be compared directly to your observations (https://www.cesm.ucar.edu/working-groups/paleo/simulations/cesm1-itrace, contact person Chengfei He, cxh1079@earth.miami.edu). I agree with the reviewer that this would be a valuable addition to the manuscript. However, it would represent a large revision. While I would be supportive of adding such a model component, this is not a requirement for publication and I am happy to consider a revised manuscript that does not include it - provided you address the other reviewer concerns satisfactorily.

Good luck with the revisions, and I look forward to seeing your response. Please let me know if you have any questions.

Best regards, Christo Buizert (CP editor)

*Dear Christo,*

*Thank you very much for inviting us to re-submit a revised version of our manuscript. We responded to all comments and suggestions by the reviewer. We agree that comparing TraCE or even i-TraCE outputs with our results would be very interesting but you are right that this would be a drastic revision. It would probably mean a drastic reorganization of the manuscript and probably would complicate and lengthen the story unnecessarily. As we also write in our response letter to the reviewer, such a comparison would probably provide enough material for a follow-up study and second manuscript. We decided to stick to the interpretation of our biomarker records as it already has several topics to discuss, i.e. the provenance and climate.*

*Kindest regards,*

*Vera Meyer and co-authors*

Reviewer 1:

The revised version of the manuscript is slightly improved; however, it is not publishable yet, and the interpretation of the results needs to be rewritten based on a more accurate analysis of model outputs. It appears that the authors struggle with atmospheric dynamics and their implications in interpreting their proxy record, blinding referring to a study (without making sure their interpretation was correct) and then also contradicting themselves (colder climate moves the jet northward as well as a warmer climate). Moreover, I still see that the authors stated their conclusions as if they were facts; instead, most of their conclusions are based on hypothetical reasonings and speculation that were not substantiated. Furthermore, I still found some improper

referencing and incorrect understanding of the NAO impacts on the Mediterranean climate. All authors should proofread the article before submitting it rather than expect the reviewers to do that on their behalf! Simple software (like Grammarly or even Word itself) would have corrected many of the typos I found). I felt that the senior authors of this work should be more involved in the writing and proofing of the manuscript and its references.

*Dear reviewer,*

*Thank you for your comments and suggestions to improve our manuscript. We addressed your concerns point by point below. Of course, all of the authors were involved in the writing and proofreading of the manuscript as well as the discussion of the data. Obviously, some typos and missing commas were not identified. For the new version, we paid extra attention to that issue.*

*Kind regards,*

*Vera Meyer and co-authors*

• Regarding the climate dynamics aspects, I identified two main issues:
1. LL164-165"Warm and wet winters in the Mediterranean are generally associated with negative NAO-states while cold and dry winters occur during positive phases (Eshel and Farrel, 2000)."
Also in L661
Again, this is not correct! Only the southern part of the Mediterranean Sea will experience warm winters with NAO-, in particular northern African countries, while the rest of the Mediterranean will see average (central part) or below average (northern part) temperature.

*We specified the region saying "southern Mediterranean" (lines 164-165; 661).*

2. LL565-566 "According to Li et al. (2019), a weakening of the Atlantic Meridional Overturning Circulation (AMOC) during HS1 would push the westerlies northward."
In the conclusions, the authors wrote:
"First the climate became arid due to a northward shift of the storm track. At 16 ka BP wet conditions established as the storm track was pushed south. Afterwards the climate remained relatively dry until 11 ka BP as the westerly jet moved north in response to ice sheet retreat." (commas are again missing after First, At 16 ka BP,... Afterwards,)

It is unfortunate that Li et al. do say that, but it seems their interpretation of the model output (their figure 7) is not correct. Figure 7a shows a strengthening and narrowing of the zonal "jet" (although 500 hPa is not the geopotential height to look at the jet stream, it should be 200 hPa!), rather than a northward shift! This is not conducive to a positive NAO phase. Moreover, at higher altitudes, the easterly winds seem to increase, which is the opposite of an NAO+. Figure 9a [max (PI) – min (EH) precession] instead shows a northward shift of the zonal winds (I wonder at what elevation as the changes are tiny); NAO+ anomalies at 500 hPa (Fig. 9b, I assume it's at 500 hPa as the authors don't mention it, and also here we are talking about changes of max 1-3 dm [if the units they said (m2/s2) are indeed what they plot), which is also tiny], and a cooling of the Arctic (that could be dynamical simply because of less meridional exchange due to NAO+).
In another paper from Löfverström and Lora (2017) (https://doi.org/10.1002/2017GL074274), both atmospheric dynamicists looking at Trace simulations (the same as Liu et al., 2019) clearly show that the weakening of the AMOC, during the H1 leads to a narrow (due to the expansion of sea-ice, that's why the jet cannot go northward!) and stronger jet (see their figure 2). Also, the jet is less variable compared to warmer climates where the jet is more tilted, broader and weaker. See also an older

paper by Li and Battisti 2008 (DOI: 10.1175/2007JCLI2166.1).

Finally, the authors, with their interpretation based on Liu et al. 2019, were contradicting themselves as for the H1 with more sea ice and colder climate, the jet somehow was moving northward, but then when the ice sheet and sea ice retreated, the jet also moved northward!

Please re-interpret your data based on what actually the models are showing. If possible, I suggest using TraCE and checking whether a shift northward or southward impacts precipitation in northeastern Africa. Another option (or additional option) could even be to use ERA5 or observations and look at what happens at precipitation when the jet is northward and when the jet is southward. For the H1, if I recall correctly, the models, in general, show an overall aridification, so I am not sure why the authors' record shows 2 phases with the second wet; maybe because in the second phase, it is getting warmer? But then why would it get dryer again till 11 kyrs ago?

*We were not aware of the controversy regarding the interpretation of the model output for HS1 in Liu et al. (2019) since none of us is an expert in paleo-climate modeling. Therefore, we are grateful to the reviewer for pointing this out. However, apart from the model, Li et al use grain size analysis of loess sequences in Central Asia to infer dynamics of the westerlies. These records suggest a stronger influence of the westerly jet at their study sites during Heinrich Stadials, which also supported their hypothesis of a northward movement. Besides, the review paper by Naughton et al. (2023) invokes northward and southward migrations of the polar front and associated Jet during HS1 to explain the two-phase pattern of HS1 described based on proxy data. In that data-based context, our inference regarding the movements of the polar front during HS1 seemed reasonable. However, considering other studies, this interpretation may be challenged. As we recall from the literature review, the response of the westerlies to AMOC slowdown during H stadials is debated. There are studies invoking a northward shift (Li et al., 2019) but others suggest the opposite (Nagashima et al., 2011). A recent study suggested that the jet axis remained stable but that the wave train was altered (Gai et al., 2023). In turn other studies indicated that the jet narrowed and strengthened at these times (e.g. Löfverström and Lora). So, just mentioning the hypothesis of the northward movement which, according to the reviewer, can be challenged from a modeler perspective, is probably not enough and one should also consider other hypotheses. At this point we agree with the reviewer and Editor that a detailed investigation of the development of the polar front in TraCE simulation and a comparison to proxy data in the Mediterranean-Eurasian realm would be a valuable and interesting endeavor to investigate whether the dynamics of the polar front can explain the hydrologic pattern in our record. However, we feel that this would be too much information for this manuscript as it would drastically lengthen it and the interpretations based on data and model might be rather complex. Such a comparison would probably be enough material for a follow-up study and publication. Therefore, we prefer to focus on the proxy data in this manuscript.*

*As a detailed investigation of the response of the jet to AMOC forcing is not the major focus of our manuscript, we decided to remove or rephrase sentences where shifts were mentioned. We talk about stronger or weaker influence of the westerlies in the new version. This leaves space for other dynamics than latitudinal movements of the jet axis including widening and narrowing of the westerly belt as well as strengthening or weakening of the winds as well changes in the wave pattern.*

*The following changes were made:*

*lines 21, 563-573 and 588-590: removed*

*513-516; 780-785: rephrased.*

*Gai, C., Wu, J., Roberts, A. P., Heslop, D., Rohling, E. J., Shi, Z., Liu, J., Zhong, Y., Liu, Y. and Liu, Q.: Heterogenous westerly shifts linked to Atlantic meridional overturning circulation slowdowns, Commun. Earth Environ., 4(1), 1–9, doi:10.1038/s43247-023-00987-z, 2023.*

*Nagashima, K. et al. Millennial-scale oscillations of the westerly jet path during the last glacial period. J. Asian Earth Sci. 40, 1214–1220 (2011).*

Things are not as straightforward and clear as the authors make them seem to be! That's why the authors should always use conditional tenses when making those speculations and also refer to studies in which climate dynamicists and modellers are interpreting model outputs.
Here is a suggestion for one point in the conclusion (see also suggested correction in capital letters)

"Amplified winter precipitation MAY HAVE resulted from a combination of regional and large-scale factors. ENHANCED Local cyclogenesis (Cyprus Lows) MAY HAVE BEEN promoted by elevated HIGHER SSTs and southward displaced westerlies (negative NAO-like pattern) steering Atlantic storms into the Mediterranean AND intensifYING the moisture delivery and precipitation in the Nile-River delta region."

*We added conditional tenses in the discussion and conclusion. Examples: lines 780-789; 736; 710.*

• I lost count of how many times the authors mentioned in the text that the increase in precipitation during the AHP over northeastern Africa is due to an increase in winter precipitation, as the summer monsoon most likely did not go further north than 22 °N. Please be concise!

*We shortened several paragraphs in the introduction and discussion to be more concise. Examples: lines 55-65; 625-639; 680-688; 691-693; 699-701.*

*We also reorganized section 5.1.*

Additional comments

The authors said in their answer that they had corrected all instances, but I still found several places where they refer to North Africa rather than northern Africa: e.g., L 29, L32, L77, ...
*The issues mentioned were changed.*

Before "which" it usually goes a comma, e.g. L99. Again, the authors should use at least a software that corrects for such basic things (even Word would do it!).
L96 comma after "basin"...

*L96: Changed. We searched for missing commas in the manuscript and added them.*

L159 "Next to the Cyprus Lows the Red Sea Trough"... comma after "Lows"
there are so many of those instances that I cannot write them all here! Please do your job to proofread it and use the appropriate software to catch those mistakes.

*Changed.*

L143 change to read "along the coast".

*Changed.*

L246 "The concentrations of HMW n-alkanoic acids and HMW n-alkanes are shown in Figure 3." What is the point of such types of sentences in the main text? Just directly describe the results as done in the following sentence and then refer to the figure.

*Sentences removed.*

LL380-381 ". Bowen et al. (2005) define the growing season as months with mean temperatures >0°C." The authors cited Bowen et al. 2005 "Global application of stable hydrogen and oxygen isotopes to wildlife forensics" for the definition of growing season. This paper clearly does not seem to really focus on plants' growing season. I already asked the authors to check their references! Anyway, that definition makes no sense; most of central and southern Europe will have a growing season of 365 days or almost! One definition that is used as an indicator of the length of the growing season (https://www.epa.gov/climate-indicators/climate-change-indicators-length-growing-season) is the interval between the last and first frost, but there are others more accurate. This won't change the authors' conclusion or approach but further shows that senior authors should be more involved in the writing process. Otherwise, such mistakes cast doubts on the whole work.

*We cited Bowen's dataset correctly and therefore did not change anything. Bowen et al., defined the growing season as days above 0°C and called their dataset dD of the growing season. We intended to correctly cite this published dataset by using its original name. We also state that in case for the Nile-River watershed this definition implies that the growing season equals the annual mean. This is also acknowledged by Bowen et al themselves with respect to the tropical regions. This is the information that matters as the annual mean is also what the plants capture in their isotopic signature. So, the dataset is applicable to our source apportionment approach, no matter if there are other definitions of the growing season that are incompatible with Bowen et al. By stating that the growing season definition implies that we are dealing with annual means is enough information for the readers to correctly understand how Bowen's dataset should be interpreted in our case. So, we do not understand why the reviewer has concerns about how carefully we checked the reference in this case. We cannot recon any misrepresentation here.*

L532 the authors mean increase in evaporation.

*Thanks for pointing out. Sentence removed to be more concise.*

L437 L443 (and other places) either use the acronym EM or not (make sure to define it the first time).

*We removed the acronym and work the full term throughout the new version of the manuscript. Figure 5 is an exception. We keep the acronym here to save space. In the caption the acronym is explained (line 581).*

LL625-628 Again, this representation of the literature is incorrect!
First of all, if the models were showing 24°N and the proxy 23-28°N, there would be a good agreement.
Second, the authors cited modelling experiments in which the Sahara is vegetated, saying the models show the monsoon goes up to 24°N; however, Pausata et al.'s study is mentioned in the first sentence when the authors say the monsoon goes up only up to 24°N, but also, in the following one when they stated it goes up to 31°N). So, to my understanding, based on what the authors wrote, the models provide a range from 24 to 31°N and proxy records from 23 to 28°N, so it would seem a good agreement. Actually, the models seem even to overestimate the proxy values...

The confusion probably arose from the fact that PMIP simulations significantly underestimate the monsoon extension, if I recall correctly, doesn't even reach 18°N ... probably 16°N. However, those simulations should not be used to compare with proxies as they are missing necessary boundary conditions (such as vegetation changes). Hence, it would be a pointless exercise.

So please try to tell the story well, referring correctly to the literature; otherwise, the paper will lose credibility.

*We are not entirely sure what the reviewer's point is. We are aware of the fact that PMIP-simulations do not include vegetation and dust feedbacks and that in many of these simulations the monsoonal extent is somewhere between 16-20°N. This underestimates what pollen and leaf wax data suggest. We did not cite the PMIP simulations along the lines and therefore do not understand why the reviewer is speculating about a confusion stemming from these simulations. We cite studies working with land-surface feedbacks which set the northernmost extent between 23-27°N. So yes, at least for some simulations, there is a relatively good match with pollen data as far as the northernmost extent of the monsoon is concerned. However, uncertainty remains regarding the consistency of models and data between ~27-31°N, where leaf-wax lipids indicate an increase in rainfall that cannot be reproduced by the majority of models (e.g. Chandan and Peltier, 2020; Thompson et al., 2019;2021). To our knowledge, there is only one simulation in Pausata et al which sets the boundary as far north as 31°N. Researchers speculate about other mechanisms driving the isotopic signal in the northern Sahara and the view of the monsoon fringe extending up to 31°N has recently been challenged by data from Morocco (Cheddadi et al., 2021). This is mentioned in lines 630-632.*

*As for the reviewers comment on the Pausata et al. citation, we cited the study in these two sentences as Pausata et al., provide a range of northernmost extents, which depend on different combinations of vegetation and dust feedbacks. The range is 26-31°N*

*For the new version, we removed the paragraph encompassing lines 625-639, which to a large extent, address the situation over northwestern Africa. We wanted to address the reviewer's recommendation of being more concise. Since the lines, are not relevant for the study site in the northeastern Sahara, we decided to remove them. The remaining paragraph focuses on northeastern Africa and the Nile delta region only. It is concluded that models and proxies agree that the monsoon fringe did not reach the Nile delta during the AHP. This is the important information.*

L647 correct "north easternmost"
*Changed to: "northeasternmost"*

L650 "...most likely was responsible for the rain ..." it should be "was most likely responsible"; again, using basic grammar correction tools would have sufficed!

*Changed.*

LL655-658 "The warm SST would have enhanced evaporation over the Mediterranean Sea ... Additionally, warming would have strengthened the formation of Cyprus Lows over the Levantine Basin." The first part should be "warm SST enhanced" as it is a fact, whereas the second is hypothetical, so it should be "may have"... as warmer SST, if accompanied by more barotropic conditions, would not lead to more precipitation at all, but drier conditions.

*Changed.*

LL701-702 The verb is missing in that sentence!

*Sentence says now: "Savannah was centred between 20-25°N partly reaching to ~28-31°N (Larrasoaña et al., 2013; Hély et al,. 2014; Giraudi et al., 2013; Hamdan et al., 2016)."*

L709-710 avoid repeating the same word in the same sentence (enhanced).

*Done.*